# Tissue-autonomous immune response regulates stress signaling during hypertrophy

**Robert Krautz[†‡*], Dilan Khalili[†], Ulrich Theopold***

Department of Molecular Biosciences, The Wenner-Gren Institute (MBW), Stockholm University, Stockholm, Sweden

**Abstract** Postmitotic tissues are incapable of replacing damaged cells through proliferation, but need to rely on buffering mechanisms to prevent tissue disintegration. By constitutively activating the Ras/MAPK-pathway via $Ras^{V12}$-overexpression in the postmitotic salivary glands (SGs) of *Drosophila* larvae, we overrode the glands adaptability to growth signals and induced hypertrophy. The accompanied loss of tissue integrity, recognition by cellular immunity, and cell death are all buffered by blocking stress signaling through a genuine tissue-autonomous immune response. This novel, spatio-temporally tightly regulated mechanism relies on the inhibition of a feedback-loop in the JNK-pathway by the immune effector and antimicrobial peptide Drosomycin. While this interaction might allow growing SGs to cope with temporary stress, continuous Drosomycin expression in $Ras^{V12}$-glands favors unrestricted hypertrophy. These findings indicate the necessity to refine therapeutic approaches that stimulate immune responses by acknowledging their possible, detrimental effects in damaged or stressed tissues.

**\*For correspondence:**
rkrautz@binf.ku.dk (RK);
uli.theopold@su.se (UT)

[†]These authors contributed equally to this work

**Present address:** [‡]The Bioinformatics Centre, Department of Biology, University of Copenhagen, Copenhagen, Europe

**Competing interests:** The authors declare that no competing interests exist.

## Introduction

Immune and stress responses have evolved to protect the organism from both exogenous and endogenous stimuli (*Eming, 2014*; *Adamo, 2017*; *Rankin and Artis, 2018*). By sensing deviations from homeostasis and inducing compensatory mechanisms, immune and stress responses keep physiological parameters within tolerable limits (*Hoffmann and Parsons, 1991*; *Vermeulen and Loeschcke, 2007*).

Heat shock, radiation, starvation, toxic metabolites, hypoxia, or hyperoxia are all well-characterized stressors. They activate stress responses which can ultimately lead to the induction of programmed cell death (*Lowe et al., 2004*; *Loboda et al., 2016*). In *Drosophila*, the Keap1-Nrf2-, JNK-, and p38-signaling pathways are crucial for mounting these anti-stress-responses (*Stronach and Perrimon, 1999*; *Fuse and Kobayashi, 2017*). Immune responses on the other hand, like the *Drosophila* Toll and imd pathways, are typically activated by molecular structures exposed on the surfaces of pathogens (*Medzhitov and Janeway, 2002*; *Kurata, 2004*). These pathways coordinate the humoral and cellular immune system to eliminate intruding pathogens (*Lemaitre and Hoffmann, 2007*; *Buchon et al., 2014*).

Humoral immune responses in *Drosophila* are characterized by the production and secretion of large sets of effector molecules, most notably antimicrobial peptides (AMPs) like Drosomycin (Drs) (*Imler and Bulet, 2005*). AMPs not only target extrinsic threats in the form of intruding pathogens, but also react to intrinsic stimuli such as tumorigenic transformation with the possibility to induce apoptosis (*Araki et al., 2019*; *Parvy et al., 2019*). However, it remains poorly understood whether AMPs also have functions beyond promoting apoptosis when sensing and reacting to accumulating stress such as during wound healing and tumor formation.

**eLife digest** Tissues and organs work hard to maintain balance in everything from taking up nutrients to controlling their growth. Ageing, wounding, sickness, and changes in the genetic code can all alter this balance, and cause the tissue or organ to lose some of its cells. Many tissues restore this loss by dividing their remaining cells to fill in the gaps. But some – like the salivary glands of fruit fly larvae – have lost this ability.

Tissues like these rely on being able to sense and counteract problems as they arise so as to not lose their balance in the first place. The immune system and stress responses are crucial for this process. They trigger steps to correct the problem and interact with each other to find a common decision about the fate of the affected tissue. To better understand how the immune system and stress response work together, Krautz, Khalili and Theopold genetically manipulated cells in the salivary gland of fruit fly larvae. These modifications switched on signals that stimulate cells to keep growing, causing the salivary gland's tissue to slowly lose its balance and trigger the stress and immune response.

The experiments showed that while the stress response instructed the cells in the gland to die, a peptide released by the immune system called Drosomycin blocked this response and prevented the tissue from collapsing. The cells in the part of the gland not producing this immune peptide were consequently killed by the stress response. When all the cells in the salivary gland were forced to produce Drosomycin, none of the cells died and the whole tissue survived. But it also allowed the cells in the gland to grow uncontrollably, like a tumor, threatening the health of the entire organism.

Mapping the interactions between immune and stress pathways could help to fine-tune treatments that can prevent tissue damage. Fruit flies share many genetic features and molecular pathways with humans. So, the next step towards these kinds of treatments would be to screen for similar mechanisms that block stress activation in damaged human tissues. But this research carries a warning: careless activation of the immune system to protect stressed tissues could lead to uncontrolled tissue growth, and might cause more harm than good.

Apart from their described individual roles, immune and stress pathways are proposed to be either induced successively or concomitantly dependent on the level of deviation from homeostasis (*Chovatiya and Medzhitov, 2014*; *Ammeux et al., 2016*). However, detailed characterization of wound healing and tumor models in *Drosophila* revealed a more complex picture (*Park et al., 2004*; *Buchon et al., 2009*; *Meyer et al., 2014*; *Wu et al., 2015*; *Liu et al., 2015*). Accordingly, immune and stress responses often neither occur separately, nor do they follow a simple linear cascade, but rather regulate each other via context-dependent mutual crosstalk (*Wu et al., 2015*; *Liu et al., 2015*; *Fogarty et al., 2016*; *Pérez et al., 2017*). One recurring motif throughout most of these models is the central role of the stress-responsive JNK-pathway and its frequent interaction with the Toll and imd immune pathways (*Park et al., 2004*; *Rämet et al., 2002*; *Galko and Krasnow, 2004*; *Uhlirova et al., 2005*; *Igaki et al., 2006*; *Enomoto et al., 2015*; *Andersen et al., 2015*). However, while JNK-signaling can function either in a tumor-promoting, anti-apoptotic or in a tumor-suppressive, pro-apoptotic manner depending on the context, Toll- and imd-signaling have only been shown to display a tumor-suppressing, pro-apoptotic role in *Drosophila* (*Uhlirova et al., 2005*; *Igaki et al., 2006*; *Enomoto et al., 2015*; *Uhlirova and Bohmann, 2006*; *Cordero et al., 2010*; *Vidal, 2010*).

These tumor-suppressive, pro-apoptotic functions of immune responses have been well characterized and attributed to the secretion of humoral factors or the recruitment of immune cells through the systemic immune system (*Fogarty et al., 2016*; *Pérez et al., 2017*; *Babcock et al., 2008*; *Pastor-Pareja et al., 2008*; *Parisi et al., 2014*; *Hauling et al., 2014*). In addition, during clonal cell competition in imaginal discs, Toll- and imd-signaling were implicated in the elimination of less fit cell clones by inducing apoptosis (*Meyer et al., 2014*). Importantly, the selective growth disadvantage of these less fit cells is thought to be a response to systemic infection (*Germani et al., 2018*) and it remains an open question whether genuine tissue-autonomous immune responses can contribute to adaptation of growth during wound healing and tumor formation.

The larval salivary gland (SG) of *Drosophila* is a powerful system to study adaptive growth control, because growth is not completely predetermined, but modulated by the nutritional status

(*Smith and Orr-Weaver, 1991*; *Britton and Edgar, 1998*). In contrast to mitotically active tissues, growth in post-mitotic tissues like the larval SG is based on endoreplication and hypertrophy rather than on cell division (*Edgar et al., 2014*; *Orr-Weaver, 2015*). Modifying the underlying, tight growth regulation, for instance by continuous growth signaling via constitutively activated Ras/MAPK-signaling can easily lead to the accumulation of oxidative stress and DNA damage (*Mason et al., 2004*; *Bartkova et al., 2005*; *Bartkova et al., 2006*; *Di Micco et al., 2006*; *Shim et al., 2013*; *Shim, 2015*). However, the parameters defining the natural limit of growth adaptation and the buffering mechanisms in place to cope with prolonged or continuous stress remain poorly understood.

Here, we uncover a genuine tissue-autonomous immune response which directly regulates hypertrophic growth and adaptation to accumulating stress in larval SGs. By overexpressing a dominant-active form of the small GTPase Ras, $Ras^{V12}$, we induced hypertrophic growth. This activates a tissue-autonomous immune response which allows the hypertrophic gland to cope with the resulting stress through spatio-temporally regulated inhibition of the JNK-mediated stress response. We present evidence that tissue-autonomous expression of the AMP Drosomycin (Drs) is at the core of this inhibition: Drs directly interferes with JNK-signaling and inhibits JNK-dependent programmed cell death. This prevents recognition of the stressed tissue by the cellular immune response and allows survival and unrestricted growth of hypertrophic SGs.

## Results

### Local immune reaction accompanies Ras$^{V12}$-dependent hypertrophy

In order to identify buffering mechanisms that compensate for continuous stress, we made use of our previously published hypertrophy model in the SGs of *Drosophila* larvae (*Hauling et al., 2014*). We expressed a dominant-active form of Ras, $Ras^{V12}$, across the entire secretory epithelium of SGs throughout larval development by using the $Bx^{MS1096}$ enhancer trap (*Figure 1—figure supplement 1A,C* for 96/120 hr after egg deposition, AED). To further enhance $Ras^{V12}$-dependent hypertrophy, we combined $Ras^{V12}$-expression with RNAi-mediated knockdown of the cell polarity gene *lethal (2) giant larvae* (lgl; *Figure 1—figure supplement 1C–H*; *Jacob et al., 1987*; *Strand et al., 1994*). Their individual and cooperative role in tumor formation in mitotic tissues has been well characterized (*Bilder et al., 2000*; *Pagliarini and Xu, 2003*; *Herranz et al., 2016*). Cell- and tissue-morphology was assessed using Phalloidin staining (*Figure 1A*; *Figure 1—figure supplement 1A,C*) and nuclear morphology by DAPI (*Figure 1B–C*; *Figure 1—figure supplement 1C–F*).

At 96 hr AED, $Ras^{V12}$-expressing SG cells retained most of their normal morphology compared to $w^{1118}$-control glands. However, their integrity and polarity were severely disrupted at 120 hr AED (*Figure 1A*; *Figure 1—figure supplement 1C*). Nuclei of $Ras^{V12}$-glands showed a continuous increase in volume at 96 hr and 120 hr AED (1.33 fold compared to $w^{1118}$ controls at 96 hr AED; 5.66 fold at 120 hr AED; *Figure 1C*) and signs of nuclear disintegration at 120 hr AED implying the induction of programmed cell death (PCD; *Figure 1B*). Both loss of cell integrity and nuclear disintegration coincided temporarily (*Figure 1A–B*) and were exacerbated upon coexpression of $l(2)gl^{RNAi}$ (*Figure 1—figure supplement 1C–F*). These results confirm our previous findings and demonstrate that continuous growth signaling in larval SGs leads to increased organ size accompanied with additional endocycles at 96 hr AED, both hallmarks of compensatory hypertrophy (*Tamori and Deng, 2014*). Furthermore, the additional, $Ras^{V12}$-induced endoreplications without obvious effect on tissue integrity imply an adaptability to excess growth signaling, whereas the subsequent collapse of nuclear integrity and cellular polarity at 120 hr AED delineate its limitations.

To characterize the mechanisms involved in the early phase of SG growth adaptation at 96 hr AED, we performed total RNA sequencing of complete $Ras^{V12}$-expressing and $w^{1118}$-control SGs dissected at 96 hr AED prior to cellular and nuclear disintegration. The most significantly upregulated gene in $Ras^{V12}$-glands compared to their $w^{1118}$-counterpart was Ras85D itself (q-value = $6.51 \cdot 10^{-282}$), which validates the experimental set-up (*Figure 1D*). The most differentially expressed gene in turn was the AMP Drs, which indicates the activation of a local immune response in $Ras^{V12}$-glands. To evaluate this further, we employed a GFP reporter for Drs and observed strong induction in $Ras^{V12}$-glands, but not in any other larval tissue (*Figure 1E*; *Figure 1—figure supplement 1B,G,I*; *Ferrandon et al., 1998*). At 96 hr AED, the entire secretory epithelium of the SG expressed Drs with a strong tendency for increased induction in the proximal part (PP) closest to the duct (*Figure 1—*

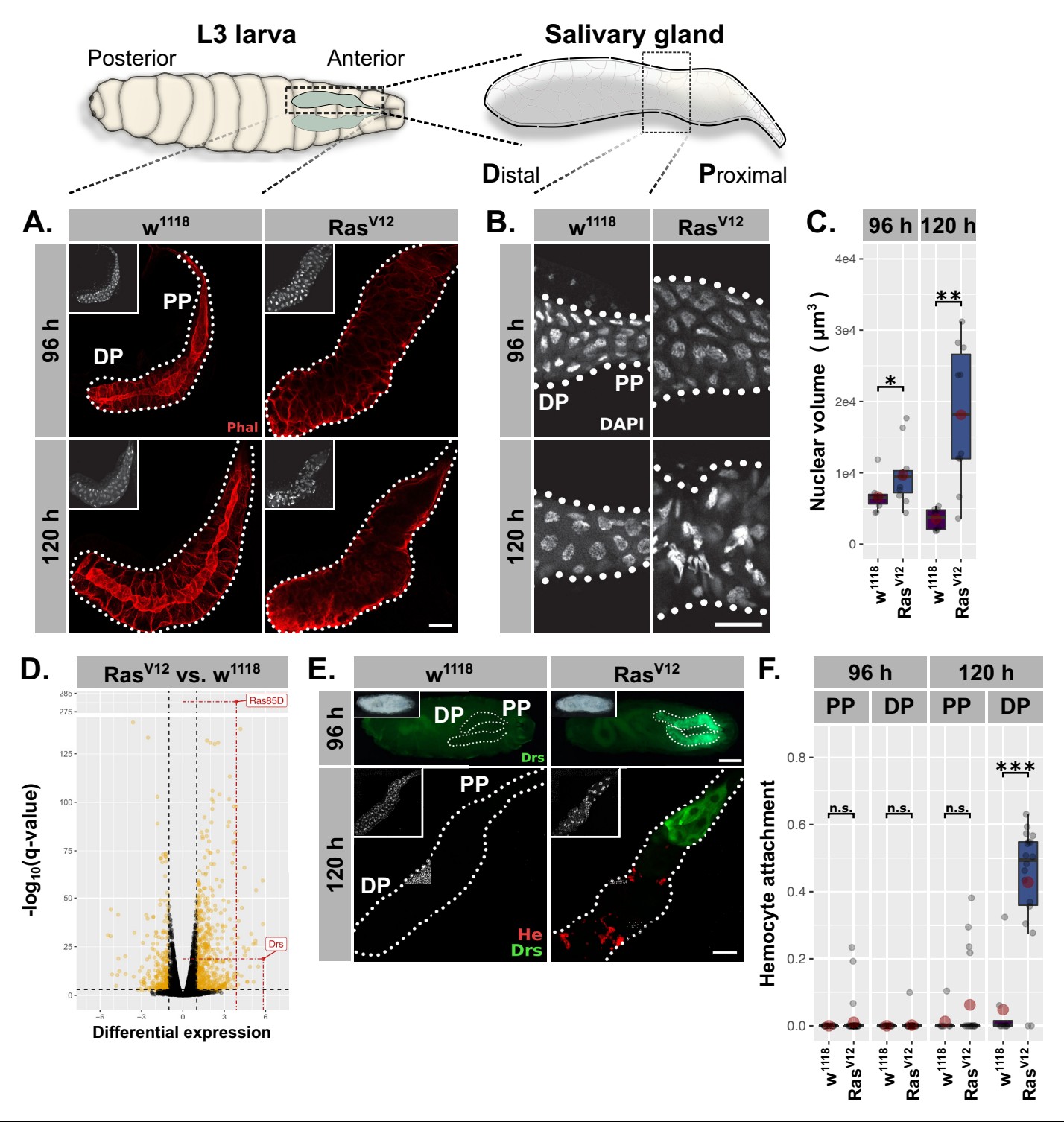

**Figure 1.** *Ras^V12*-induced hypertrophy induces local and cellular immune responses. (**A**) *Ras^V12*-glands and controls stained with Phalloidin (red) to monitor tissue integrity at 96 hr and 120 hr after egg deposition (AED). (**B**) Nuclei stained with DAPI (white) to visualize nuclear volume and disintegration at 120 hr AED in *Ras^V12*-glands. (**C**) Nuclear volume as quantified by z-stacks of DAPI-stained salivary glands (SGs) at 96 hr and 120 hr AED and averaged per gland. (**D**) Comparative transcriptome analysis of *Ras^V12*- vs. *w^1118*-glands. Differential expression quantified as beta statistic with q-values by Wald test. Significantly differentially expressed genes (*log2(beta) ≥1; q-value ≤0.05*) highlighted in yellow. (**E**) Upper: Whole larvae with DrsGFP reporter (green) expressing *Ras^V12* in glands or controls at 96 hr AED. Lower: *Ras^V12*- and control-glands with DrsGFP reporter (green) stained for hemocytes (anti-Hemese, red). Proximal and distal gland parts are indicated by 'PP' and 'DP'. (**F**) Hemocyte attachment measured as *ln(Hemese-*

*Figure 1 continued on next page*

*Figure 1 continued*

area)/ln(SG-area) and separated by time and gland part. Insets: (**A/E** Lower) DAPI, (**E** Upper) brightfield. Scalebars: (**A-B, E** Lower) 100 μm, (**E** Upper) 500 μm. Boxplots in (**C, F**): lower/upper hinges indicate 1st/3rd quartiles, whisker lengths equal 1.5*IQR, red circle and bar represent mean and median. Significance evaluated by Student's t-tests (***p<0.001, **p<0.01, *p<0.05, n.s. p≥0.05).

The online version of this article includes the following figure supplement(s) for figure 1:

**Figure supplement 1.** Homeostasis, local and cellular immune responses separate along the longitudinal axis of $Ras^{V12}$-glands.

*figure supplement 1B,G*). At 120 hr AED Drs was almost exclusively expressed in the PP, an expression pattern that persisted until $Ras^{V12}$-expressing larvae belatedly pupariate (*Figure 1E*; *Figure 1— figure supplement 1B,G,I*). To rule out artifacts derived from driver related unequal expression across the SG epithelium, we measured $Bx^{MS1096}$-driven RFP-expression across the longitudinal gland axis at 96 hr and 120 hr AED and normalized the acquired signal with driver-independent Phalloidin-fluorescence (*Figure 1—figure supplement 1A*; see 'Materials and methods'). The average Drs-reporter signal already peaks at 96 hr AED in the middle of the gland epithelium, while $Bx^{MS1096}$-driven expression increases further until 120 hr AED. Hence, while $Bx^{MS1096}$-driven expression of $Ras^{V12}$ is crucial for activating Drs-expression, both are not qualitatively correlated (*Figure 1— figure supplement 1A–B*). In order to assess whether hemocytes were recruited as part of a parallel, cellular immune response, we stained glands with an antibody against the pan-hemocyte-marker Hemese. While $Ras^{V12}$-glands were completely devoid of hemocytes at 96 hr AED, at 120 hr AED they were recruited to the gland surface. However, hemocyte attachment was restricted to the distal, non-Drs expressing part (DP), rendering Drs expression and hemocyte attachment across the gland epithelium mutually exclusive (*Figure 1E–F*). Coexpression of $l(2)gl^{RNAi}$ elevated the level of recruited hemocytes at 120 hr AED and pre-empted this recruitment to the DP already at 96 hr AED (*Figure 1—figure supplement 1G–H*).

We next investigated whether the change in nuclear volume as a marker for growth adaptation follows a similar proximal-distal-divide as Drs-expression and hemocyte attachment (*Figure 1—figure supplement 1E*). Nuclei in the DP of the SG at 96 hr AED showed a moderate volume increase upon $Ras^{V12}$-expression compared to distal $w^{1118}$-control nuclei. However, after 120 hr AED distal nuclei had undergone a drastic increase in nuclear volume (6.28 fold compared to distal $w^{1118}$-control nuclei) while nuclei in the PP of the SG displayed only a moderate increase in size compared to $w^{1118}$-control nuclei, that did not increase over time. This indicates that nuclei in the DP of $Ras^{V12}$-glands undergo more rounds of endoreplication than their proximal counterparts coinciding with the decline of Drs-expression and an increase in hemocyte attachment in this part. Moreover, this difference also explains the marginal reduction in $Bx^{MS1096}$-driven expression from DP to PP (*Figure 1— figure supplement 1A*; PP shows 86% of DP-expression level).

## Dorsal-dependent Drs expression is part of a genuine tissue-autonomous immune response

As barrier epithelia, the lumen of the SG forms part of a continuum with the exterior, exposing them to extrinsic stimuli including nutritional cues and pathogens (*Andrew et al., 2000*). Since systemic infection can modulate tissue growth, we sought to clarify whether the observed local immune response in the gland epithelium fulfills the criteria of a genuine tissue-autonomous immune response or was rather embedded in a wider systemic immune response (*Germani et al., 2018*). Therefore, we eradicated the majority of putative systemic infections, food-derived signals and pathogens or bacterial contamination by raising larvae with $Ras^{V12}$-glands under sterile conditions, on minimal medium or by bleaching embryos (*Figure 2—figure supplement 1A–A''*; *Shaukat et al., 2015*; *Kenmoku et al., 2017*; *Asri et al., 2019*). None of these changes diminished the Drs expression, strongly indicating that Drs is indeed induced in a *bona fide* tissue-autonomous manner as a response to $Ras^{V12}$-dependent hypertrophic growth (*Figure 2—figure supplement 1A–A''*; *Colombani et al., 2005*; *Mirth et al., 2014*).

We further sought to identify the upstream factors controlling Drs expression. The homeobox-like transcription factor Caudal is necessary for Drs expression in the female reproductive organs and the adult SG (*Ferrandon et al., 1998*; *Ryu et al., 2004*; *Han et al., 2004*). However, RNAi-lines directed against Caudal or its canonical interaction partner, Drifter (dfr / vvl), did not reduce Drs expression

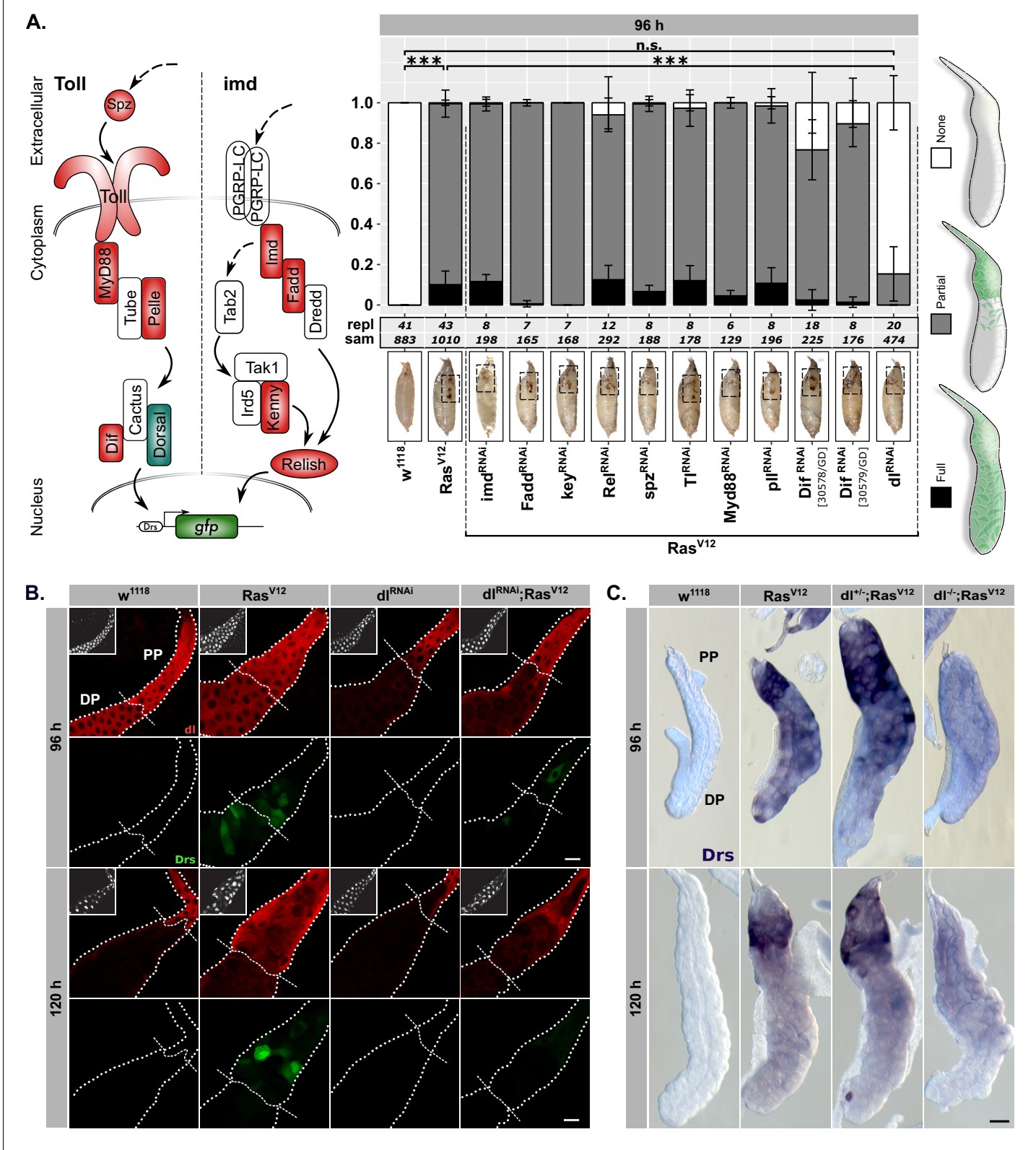

**Figure 2.** Drs expression is part of a genuine tissue-autonomous immune response. (**A**) Semi quantitative DrsGFP reporter assay to identify upstream effectors of Drs expression by RNAi in *Ras^V12*-glands. Schematic representation (left) of the Toll-/imd-pathways showing components with (green) and without an effect (red). The three distinguished phenotypes (right) were scored per replicate, means and standard deviations plotted (middle) and p-values for 'None'-phenotype calculated via Dunn's test based on Kruskal-Wallis rank sum test (***p<0.001, n.s. p≥0.05). Monitoring melanization in

*Figure 2 continued on next page*

Figure 2 continued

pupae (insets) confirmed $Ras^{V12}$-expression to avoid false positives (*Hauling et al., 2014*). (B) dl staining (red) and DrsGFP signal (green) in $Ras^{V12}$-glands with and without dl-knock-down. Dashed lines added to aid separating proximal part (PP) from distal part (DP). (C) In-situ hybridization for endogenous Drs in $Ras^{V12}$-glands hetero- or homozygous mutant for dl ($dl^{15}$). Insets: (B) DAPI. Scalebars: (B–C) 100 μm.

The online version of this article includes the following figure supplement(s) for figure 2:

**Figure supplement 1.** Drs expressed in a *bona fide,* tissue-autonomous, $Ras^{V12}$-dependent manner.

**Figure supplement 2.** Drs expression in $Ras^{V12}$-glands is dorsal-dependent.

**Figure supplement 3.** Mef2 contributes to Drs expression in $Ras^{V12}$-glands.

in $Ras^{V12}$-glands indicating that the regulation of Drs expression as part of the tissue-autonomous immune response is clearly different from its counterpart in the adult SG (*Figure 2—figure supplement 2A*; *Junell et al., 2010*).

In order to evaluate whether either Toll- or – as is the case for local infections – imd-signaling plays a role in Drs expression, we used the reproducible fluorescence pattern of the Drs-GFP reporter at 96 hr AED to assay RNAi-lines directed against canonical components of both pathways in $Ras^{V12}$-glands (*Figure 2A*; see 'Materials and methods' for scored phenotypes; *Ferrandon et al., 1998*; *Tzou et al., 2000*; *Takehana et al., 2004*; *Wagner et al., 2009*). Of the 11 tested RNAi-lines, most of which were previously published to cause phenotypes, only one targeting the NFκB-transcription factor Dorsal (dl) significantly reduced the fluorescence signal of the Drs-reporter (*Figure 2A*; *Supplementary file 3*). However, Drs expression is completely independent of the upstream modules of the two classical *Drosophila* immune pathways, Toll and imd.

The dl-Drs-relationship in $Ras^{V12}$-glands was further confirmed by the strong correlation of nuclear signals between dl-immunofluorescence and Drs-reporter intensities across PP and DP at 96 hr AED (*Figure 2—figure supplement 2B*; see 'Materials and methods'). Moreover, at 96 hr AED dl was present in the entire secretory epithelium of both $Ras^{V12}$- and $w^{1118}$-control glands. In contrast, at 120 hr AED its expression was solely confined to the PP (*Figure 2B*). This overlapped with the Drs-mRNA expression as determined by in situ hybridization (ISH; *Figure 2—figure supplement 2C*). Furthermore, both SG-specific knock-down or whole organism homozygous knock-out ($dl^{15}$) of dl abolished the majority of Drs-expression in the DP of the gland at 96 hr AED and reduced it in the PP at both time points (*Figure 2A–C*; *Figure 2—figure supplement 2C–D*). On the contrary, $Ras^{V12}$-glands hetero- or homozygous mutant for Myd88 ($Myd88^{KG03447}$) do not exhibit abolished Drs-mRNA-expression at 120 hr AED in line with our results from the reporter assay (*Figure 2—figure supplement 2E*).

To assess the extent to which Drs relies on dl for its expression, we measured the effect size on Drs expression of the employed $dl^{RNAi}$-construct in $Ras^{V12}$-glands (*Figure 2—figure supplement 3A*). A reduction of Drs-expression to 6% in the DP of 96-hr-old $dl^{RNAi};Ras^{V12}$- compared to $Ras^{V12}$-glands indicates not only a strong reliance for Drs on dl in this part at 96 hr, but also a high efficiency of the $dl^{RNAi}$-construct. However, a less pronounced decrease of Drs-expression in the PP at 96 hr and 120 hr AED to 27% and 52% upon $dl^{RNAi}$-coexpression implies the presence of additional Drs-regulating factors. Putative candidates for this Drs-regulating role were identified by screening transcription factor binding sites amongst the up- and downregulated genes in $Ras^{V12}$-glands and all other acquired transcriptomes (*Figure 2—figure supplement 3B*; see *Figure 3* for $lgl^{RNAi};Ras^{V12}$, $lgl^{RNAi};Ras^{V12}$ – PP). Of the six identified candidates, only one RNAi-line targeting the transcription factor Mef2 significantly reduces the expression of Drs as assayed by DrsGFP-reporter signal at 96 hr AED (*Figure 2—figure supplement 3C*). This is further emphasised by clusters of putative Mef2-binding sites just downstream of the *Drs*-locus and in the 5'-ends of the *Dif*- and *dl*-loci, implying both direct and indirect regulatory potential on Drs-expression (*Figure 2—figure supplement 3D*).

Together, our results indicate that the spatio-temporal dynamics of Drs expression are strongly correlated to the decrease in endogenous dl expression, independent of canonical Toll- and imd-signaling. While dl expression is $Ras^{V12}$-independent, Drs is only expressed in the presence of dl during $Ras^{V12}$-induced hypertrophy. Alongside dl, Mef2 regulates Drs-activation emphasising the notion of non-canonical activation of this immune effector even further.

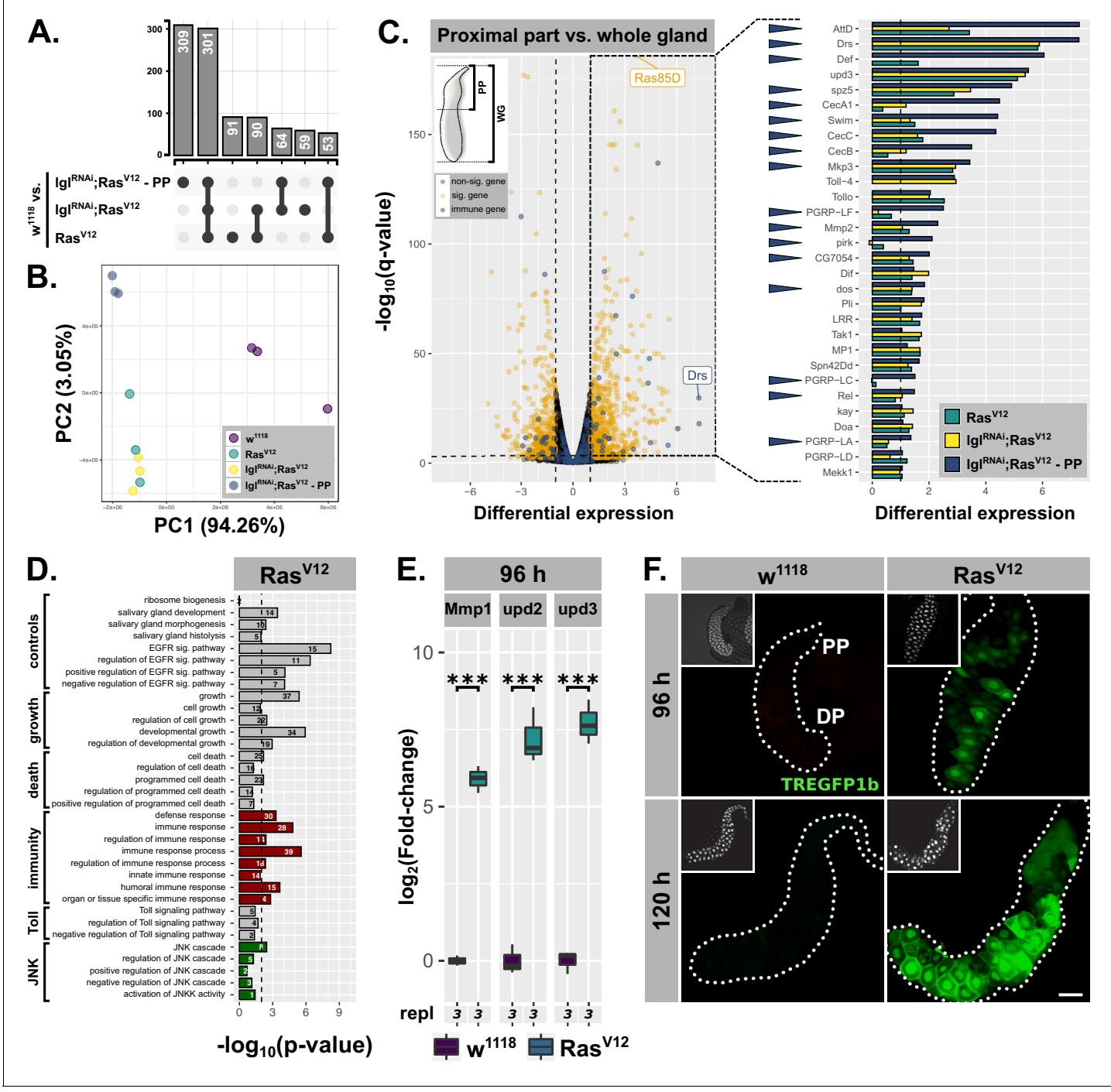

**Figure 3.** Hypertrophic *Ras^V12*-glands induce parallel immune and stress responses. (A) Common and specific genesets significantly upregulated (*log2 (beta) ≥1; q-value ≤0.05*) in either *Ras^V12*, *Igl^RNAi;Ras^V12*, or *Igl^RNAi;Ras^V12*-PP compared to *w^1118*-glands. (B) PCA including all transcriptome replicates of all sequenced genotypes. (C) Left: Comparative transcriptome analysis between proximal part (PP) of *Igl^RNAi;Ras^V12*- and entire *w^1118*-glands. Significantly differentially expressed genes (*log2(beta) ≥1; q-value ≤0.05*) and genes belonging to GO-term 'immune response' (GO:0006955) highlighted in yellow and blue. Right: Gene expression in *Ras^V12*, *Igl^RNAi;Ras^V12* or *Igl^RNAi;Ras^V12*-PP compared to *w^1118*-glands for immune genes significantly upregulated in the PP. Blue arrows indicate strongest expression in the PP for the indicated genes between all three groups. Missing bars indicate absence of expression values in the RNAseq data. (D) GO term enrichment among significantly upregulated genes in *Ras^V12*-glands including terms related to activation of JNK (green) and immune responses (red). Numbers in bars indicate amount of upregulated genes belonging to associated GO term. (E) qPCR results for canonical JNK target genes (*log2*-transformed, fold-change over *Rpl32*) at 96 hr after egg deposition (AED). Lower/upper hinges of boxplots indicate 1st/3rd quartiles, whisker lengths equal 1.5*IQR and bar represents median. Significance evaluated by Student's t-tests (***p<0.001). (F) TREGFP1b reporter (green) signal in *Ras^V12*- and control-glands at 96 hr and 120 hr AED. Scalebar: 100 μm.

*Figure 3 continued on next page*

*Figure 3 continued*

The online version of this article includes the following figure supplement(s) for figure 3:

**Figure supplement 1.** Drs is the only AMP detected to be expressed in the distal part of $Ras^{V12}$-glands until 96 hr.

**Figure supplement 2.** JNK-signaling is predominantly activated in the distal part of $Ras^{V12}$-glands.

## Hypertrophy in SGs induces parallel immune and stress responses

Between 96 hr and 120 hr AED, the decrease in Drs-expression (*Figure 1E*; *Figure 2B–C*) is correlated with deterioration of tissue integrity (*Figure 1A–B*) in the DP of hypertrophic $Ras^{V12}$-glands. Thus, we hypothesized that the tissue-autonomous immune response revealed by Drs-expression aids in preventing the collapse of nuclear as well as cellular integrity until 96 hr AED in the DP and due to its prolonged expression in the PP until 120 hr AED and beyond (*Figure 1—figure supplement 1I*).

To identify possible targets and effector mechanisms of the immune response that buffer the detrimental effects of hypertrophic growth, we analyzed the transcriptome data acquired for the $w^{1118}$-control and $Ras^{V12}$-glands at 96 hr AED in further detail. In order to distinguish whether the differences between $Ras^{V12}$- and $lgl^{RNAi};Ras^{V12}$-glands (*Figure 1—figure supplement 1C–H*) are of quantitative or qualitative nature and to characterize the PP with its persistent dl and Drs expression in depth, we also profiled transcriptomes of entire $lgl^{RNAi};Ras^{V12}$-glands and solely the PP of $lgl^{RNAi};Ras^{V12}$-glands at 96 hr AED (*Figure 3—figure supplement 1A*; see 'Materials and methods').

The sets of significant and differentially upregulated genes (i.e. *log2(beta)* $\geq$ 1; *q-value* $\leq$ 0.05) for all three conditions (i.e. $Ras^{V12}$ / $lgl^{RNAi};Ras^{V12}$ / $lgl^{RNAi};Ras^{V12}$-PP each normalized to $w^{1118}$-controls) were intersected to determine common and specific genesets (*Figure 3A*). Notably, the two biggest genesets are differentially upregulated genes shared between all three conditions and genes specifically upregulated in the PP (*Figure 3A*). Furthermore, while all 6 $Ras^{V12}$- and $lgl^{RNAi};Ras^{V12}$-replicates are in close proximity along the first two principle components in the PCA, the sets of replicates for $w^{1118}$ and especially the PP are distant from the rest along PC2 emphasizing the distinctiveness of the PP compared to the rest of the gland (*Figure 3B*).

By screening for enriched gene ontology (GO) terms amongst the significantly upregulated genes in the PP of $lgl^{RNAi};Ras^{V12}$-glands in comparison to the entire $w^{1118}$-glands, we identified genes belonging to the GO-term 'immune response' (i.e. GO:0006955) as significantly enriched (*Figure 3C*; p-value=1.45·10$^{-4}$). Detailed examination of the fold-enrichment of these genes across all three experimental groups (i.e. $Ras^{V12}$, $lgl^{RNAi};Ras^{V12}$, $lgl^{RNAi};Ras^{V12}$-PP) showed a preferential expression in the PP, too (17 of 30 genes highest expressed in PP; blue arrowheads in *Figure 3C*). In addition, 5 of the top 20 upregulated genes in the PP belong to this GO-term as well. Thus, the Drs-expression we observed using reporter lines serves as a proxy for a more complex, tissue-autonomous immune response in hypertrophic glands, especially in the PP. Nonetheless, Drs itself remains one of the top significantly, differentially upregulated genes across all three conditions (*Figure 1D*; *Figure 3C*; not shown for $lgl^{RNAi};Ras^{V12}$). In fact, Drs-expression in the PP compared to the two conditions for entire glands is even further increased, confirming the strong tendency for proximal over distal Drs-GFP reporter activation (*Figure 1E*; *Figure 2B*; *Figure 3C*). Moreover, AttD and Drs are the only two AMPs that showed considerable expression in the two groups involving whole gland samples (i.e. $Ras^{V12}$, $lgl^{RNAi};Ras^{V12}$). Strong expression of particular genes in the PP might occur as expression throughout the whole gland samples. Therefore, we tested AttD expression separately in PP and DP, which showed almost exclusive AttD-expression in the PP at 96 hr and 120 hr AED (*Figure 3—figure supplement 1D*). Similarly, CecA1 as the 4$^{th}$ highest expressed AMP in the PP-RNA-seq was also confined exclusively to the PP as evaluated by ISH (*Figure 3—figure supplement 1C, E*). Thus, based on the RNAseq and additional evaluations, Drs appears to be the only AMP to be significantly expressed in the DP until 96 hr AED, while in parallel the PP shows a much broader immunocompetence.

A similar GO-term analysis amongst significantly, upregulated genes in entire $Ras^{V12}$- or $lgl^{RNAi};Ras^{V12}$-glands revealed the enrichment of genes associated with GO-terms related to 'growth', 'salivary gland development' and 'EGFR signaling' consistent with the studied tissue and the induced $Ras^{V12}$-overexpression. Amongst the genes enriched in both $Ras^{V12}$- and $lgl^{RNAi};Ras^{V12}$-glands were also sets significantly overlapping with GO-terms indicating activated JAK-STAT-signaling (data not

shown). By using a construct reporting Stat92E-expression, we were able to confirm this geneset enrichment and show restriction of Stat92E-expression to the PP throughout development of $Ras^{V12}$-glands (*Figure 3—figure supplement 1F*; *Bach et al., 2007*). This may implicate JAK-STAT-signaling in the maintenance of immune competence in the PP and underlines the separation into PP and DP along the gland epithelium further. Moreover, the lack of unique, significantly enriched GO terms and thus distinct gene expression signatures for either $Ras^{V12}$- or $lgl^{RNAi};Ras^{V12}$-glands at 96 hr AED excludes qualitative differences between these two genotypes as an explanation for their phenotypic differences (*Figure 3—figure supplement 1B*).

Importantly, we detected signatures of an activated JNK-cascade as well as cell death in $Ras^{V12}$- and $lgl^{RNAi};Ras^{V12}$-transcriptomes indicating the presence of an activated stress response and confirming the stimulation of PCD as implied by nuclear disintegration at 120 hr AED (*Figure 1B*; *Figure 3D*; *Figure 3—figure supplement 1B*; GO:0006955; GO:0008219). To validate these signatures further, we performed qPCR for canonical JNK-targets at 96 hr AED and found that the expression of all tested genes was significantly increased compared to $w^{1118}$-control glands (*Figure 3E*; *Figure 3—figure supplement 2A*). We used the TRE-GFP1b-reporter construct, which recapitulates JNK-activation by expressing GFP under the control of binding sites for JNK-specific AP-1 transcription factors (*Figure 3F*; *Chatterjee and Bohmann, 2012*). This not only confirmed JNK-signaling in $Ras^{V12}$-glands, but also uncovered its prevalence in the DP of these glands at 96 hr and 120 hr AED consistent with the transcriptome data (*Figure 3F*; *Figure 3—figure supplement 2B*). Moreover, the elevated fluorescence signal in the DP at 120 hr AED compared to 96 hr AED implies an increase in activation over time correlating with the collapse of nuclear and cellular integrity during this period.

In summary, the transcriptome analysis confirms our findings that $Ras^{V12}$-overexpression induces a strong tissue-autonomous immune response in the PP of the SG, beyond sole Drs expression. In contrast, the DP shows a striking increase in JNK-signaling which correlates with decreasing Drs expression and cellular and nuclear disintegration at 120 hr AED, consistent with the described role of JNK target genes in PCD.

## Drs overexpression and JNK inhibition prevent $Ras^{V12}$-induced tissue disintegration

The increase in JNK-signaling in the DP of the SG as revealed by our transcriptome analysis coincided in space and time with the downregulation of Drs suggesting an interaction between the tissue-autonomous immune response and the stress response.

To test this assumption, we first overexpressed either Drs or a dominant negative form of the *Drosophila* Jun kinase jnk (basket) individually with $Ras^{V12}$ throughout the entire secretory epithelium of the SG (*Figure 4A*). Either Drs overexpression or JNK inhibition had a profound effect on the gland size, which was significantly increased at 120 hr AED, compared to $Ras^{V12}$ or UAS-dilution control $mCD8::RFP;;Ras^{V12}$ only (*Figure 4B–C*; *Figure 4—figure supplement 1A*).

Importantly, despite this size increase, glands of both genotypes (i.e. $Drs,Ras^{V12}$ / $jnk^{DN};;Ras^{V12}$) showed no morphological abnormalities and resembled control $w^{1118}$- much more than $Ras^{V12}$-glands (*Figure 4A–B*). Strikingly, $jnk^{DN};;Ras^{V12}$- and $Drs,Ras^{V12}$-glands were completely devoid of attached hemocytes (*Figure 4B,D*). Given the basement membrane's (BM) role in directly regulating organ morphology, the rescue of the gland shape upon coexpression of either Drs or jnk$^{DN}$ with $Ras^{V12}$ pointed towards changes in the integrity of the BM (*Ramos-Lewis and Page-McCaw, 2019*). In addition, previous reports suggested that hemocytes are only recruited to tissue surfaces upon tissue disintegration and when the integrity of the BM is lost (*Kim and Choe, 2014*). To trace the BM we used an endogenous GFP-trap in the *viking* gene, which encodes one subunit of CollagenIV (*Figure 4B*). Both, in the presence of Drs or by inhibiting jnk, the BM remained a continuous sheet surrounding the entire gland, whereas the BM on the surface of $Ras^{V12}$-glands was clearly disrupted.

Matrix metalloproteinases (Mmps) are likely candidates for executing the disruption of the BM and known target genes of the JNK-pathway (*Uhlirova and Bohmann, 2006*; *Hauling et al., 2014*; *Stevens and Page-McCaw, 2012*). In order to see whether hypertrophic glands express Mmps, we performed qPCRs for Mmp1 and Mmp2 at 96 hr and 120 hr AED on $Ras^{V12}$- and control $w^{1118}$-glands. At 96 hr AED, $Ras^{V12}$-glands already exhibit increased Mmp1 expression (*Figure 4E*). However, Mmp2 only reaches a significant level of expression at 120 hr AED coinciding with the appearance of hemocyte attachment (*Figure 4D–E*). SG-wide, $Ras^{V12}$-independent overexpression of Mmp2, but not of Mmp1, caused opening of the BM and hemocyte attachment to the surface

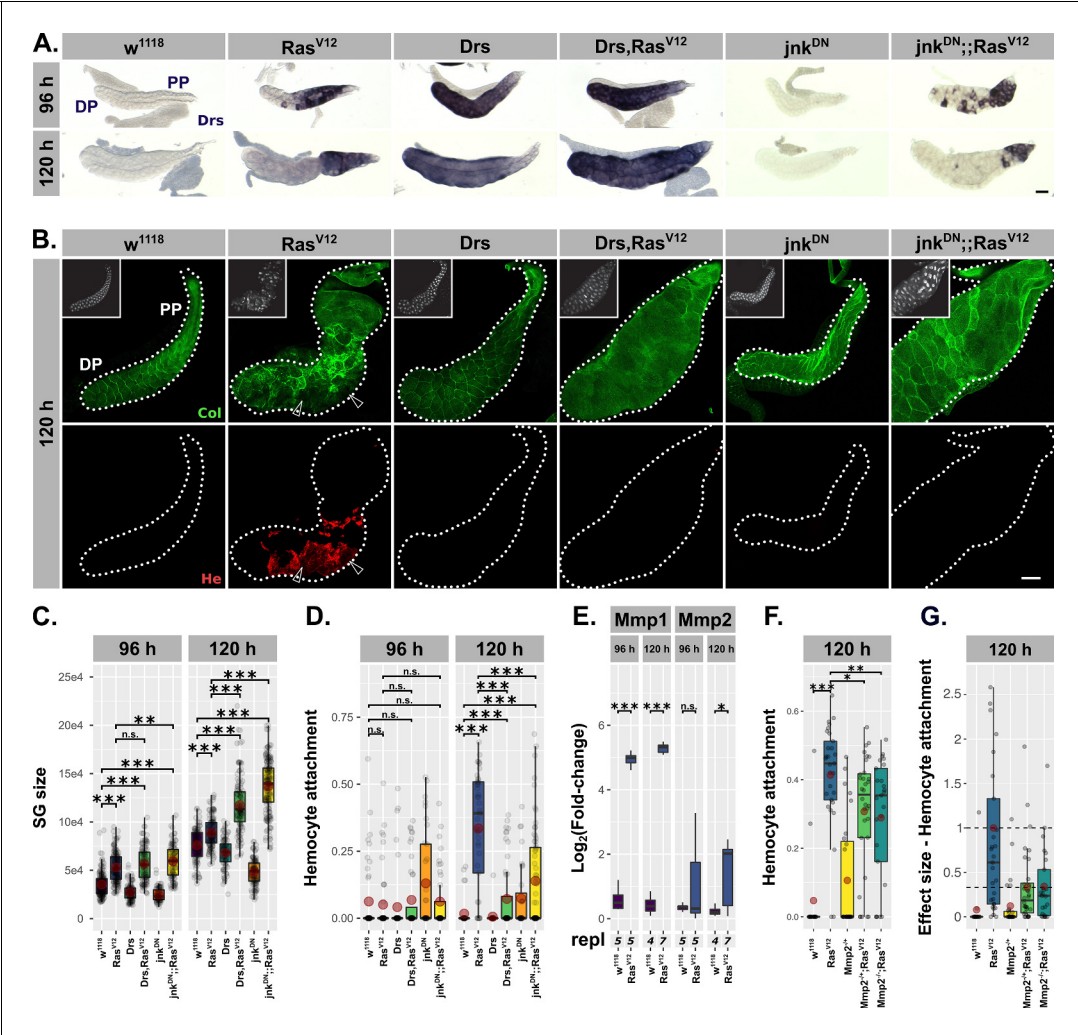

**Figure 4.** Drs overexpression and JNK inhibition individually prevent tissue disintegration. (**A**) Drs-specific in-situ hybridization identifies endogenous (*Ras^V12*, *jnk^DN*;;*Ras^V12*) and exogenous (*Drs*, *Drs,Ras^V12*) Drs expression. (**B**) Collagen-GFP trap (*vkg^G00454*, green) and Hemese staining (red) identify integrity of BM and hemocyte attachment to gland surface. Arrows indicate BM-free areas occupied and surrounded by hemocytes. Insets show DAPI (white) staining. (**C**) Size of salivary glands measured as outlined area in captured images of indicated genotypes at 96 hr and 120 hr after egg deposition (AED). (**D**) Hemocyte attachment at 96 hr and 120 hr AED represented as *ln(Hemese-area)/ln(SG-area)*. (**E**) *log2*-transformed, *Rpl32*-normalized gene expression values for matrix metalloproteinases obtained by qPCR at 96 hr and 120 hr AED. (**F**) Hemocyte attachment at 120 hr AED in *Ras^V12*-glands hetero- and homozygous mutant for Mmp2 (*Mmp2^k00604*). (**G**) Effect size of hetero- and homozygous mutant Mmp2-allele (*Mmp2^k00604*) on hemocyte attachment to and compared to *Ras^V12*-glands. Mean of hemocyte attachment in *Ras^V12*-glands at 120 hr was set to '1'. (**E**) and (**F**) represent the same data points. Scalebars: (**A–B**) 100 μm. Boxplots in (**C–G**): lower/upper hinges indicate 1st/3rd quartiles, whisker lengths equal 1.5*IQR, red circle and bar represent mean and median. Significance evaluated by Student's t-tests (\*\*\*p<0.001, \*\*p<0.01, \*p<0.05, n.s. p≥0.05). The online version of this article includes the following figure supplement(s) for figure 4:

**Figure supplement 1.** Hemocyte recruitment requires JNK-dependent Mmp2 expression.

(*Jia et al., 2015*). In contrast, knock-out of Mmp2 reduces hemocyte recruitment significantly compared to *Ras^V12*-glands (*Figure 4F–G*; 33% in *Mmp2^-/+;Ras^V12*; 34% in *Mmp2^-/-;Ras^V12*). In turn, residual hemocytes still attached to the gland surface of *Mmp2^-/-;Ras^V12*-glands were not activated (i.e. no filo- or lamellipodia, no spreading; *Figure 4—figure supplement 1C*). This indicates the necessity for a JNK-dependent expression of Mmp2 in the hypertrophic *Ras^V12*-glands prior to the recruitment of hemocytes to the tissue surface. However, the presence of residual hemocytes indicates additional cues such as reactive oxygen species (ROS) as recently suggested for neoplastic tumors (*Diwanji and Bergmann, 2019*). In fact, while *Ras^V12*-glands show accumulation of ROS in the DP, this accumulation is reverted upon Drs-overexpression or JNK-inhibition (*Figure 4—figure*

supplement 1D). A parallel block of ROS-accumulation and Mmp2-dependent BM-degradation in *Drs,Ras^V12* and *jnk^DN;;Ras^V12* could explain the absence of any attached hemocytes on the surface of these glands (*Figure 4B,D*).

Taken together, overexpression of Drs alone is sufficient to mimic the inhibition of the JNK-pathway in *Ras^V12*-glands: both lead to excess hypertrophic growth compared to *Ras^V12*-glands, but simultaneously prevent tissue disintegration and PCD. Prohibiting the disruption of the basal membrane reduces hemocyte recruitment and activation, thus preventing the cellular immune response from sensing hypertrophic growth. This in turn suggests that the endogenous Drs expression in *Ras^V12*-glands is seminal for maintaining nuclear and tissue integrity and thus part of the buffer mechanism to adapt to continuous growth signaling.

## Drs inhibits JNK-signaling

The strong correlation between loss of Drs and the increase in JNK-signaling in the DP of *Ras^V12*-glands between 96 hr and 120 hr AED indicated an active interaction between Drs and the JNK-pathway, which prompted us to resolve their hierarchy by epistatic analysis.

Coexpression of *jnk^DN* with *Ras^V12* did not deplete Drs in the PP of 120-hr-old glands, since neither the fluorescence signal of the Drs-GFP reporter nor staining for endogenous *Drs*-mRNA via ISH showed any effects (*Figure 4A*; *Figure 5—figure supplement 1A*). This excludes a direct regulation of Drs by JNK-signaling and also negates indirect reduction of Drs-expression in the DP due to JNK-induced PCD at 120 hr AED (see Figure 6 for more details). qPCR for Drs in *Ras^V12*- and *jnk^DN;Ras^V12*-glands confirmed these results further (*Figure 5—figure supplement 1B*). In contrast, qPCR in hypertrophic *Ras^V12*-glands showed a significant reduction in expression of JNK target genes upon coexpression of Drs at 96 hr and 120 hr AED (*Figure 5B*; *Figure 5—figure supplement 2C,D*), which could not be reciprocated by coexpressing mCD8::RFP as a UAS-dilution control (*Figure 5—figure supplement 3A*) and which was in line with a decrease in activated jnk (*Figure 5C,E*; *Figure 5—figure supplement 3B–C*) and TRE-GFP1b signal in *Drs,Ras^V12*- compared to *Ras^V12*-glands (*Figure 5B,D,F*). Thus, the overexpression of the AMP Drs actively and tissue-autonomously inhibits the JNK-dependent stress response in hypertrophic *Ras^V12*-glands beyond 96 hr AED.

In addition, we used an efficient RNAi-line against Drs, which in combination with Ras^V12 reduced Drs-expression significantly compared to its expression in *Ras^V12*-glands as shown by qPCR and ISH (*Figure 5—figure supplement 2A–D*; 8% residual Drs-expression at 96 hr and 4% at 120 hr in *Drs^RNAi;Ras^V12*- compared to *Ras^V12*-glands). All significantly upregulated JNK-target genes in *Ras^V12*-glands apart from *upd2* were further increased upon knockdown of Drs at 96 hr AED (*Figure 5—figure supplement 2C*). However, this effect ceased at 120 hr AED (*Figure 5—figure supplement 2D*) consistent with a lack of change in activated jnk (*Figure 5—figure supplement 2F,H*; *Figure 5—figure supplement 3B–C*) and TRE-GFP1b-signal (*Figure 5—figure supplement 2E,G*), as well as the canonical loss of Drs expression in the DP of *Ras^V12*-glands (*Figure 1E*). This result demonstrates that until 96 hr AED the endogenously expressed Drs has the same inhibitory effect on the JNK-pathway as the Drs-overexpression has at 120 hr AED.

## Drs prevents cell death in Ras^V12-glands

The capacity to induce apoptosis is a well-established function of the JNK-pathway in *Drosophila* (*Uhlirova et al., 2005*; *Igaki et al., 2006*; *Enomoto et al., 2015*; *Uhlirova and Bohmann, 2006*; *Cordero et al., 2010*). Combined with the identification of 'cell death' as a significantly enriched term in the GO-analysis, we hypothesized that the observed nuclear disintegration in *Ras^V12*-glands after 96 hr AED was a consequence of the JNK-dependent induction of PCD (*Figure 1B–C*; *Figure 1D–F*; *Figure 3D*; *Figure 3—figure supplement 1B*).

In mitotic tissues, the apoptotic inducer *head involution defective* (hid) is inhibited by Ras/MAPK-signaling (*Bergmann et al., 1998*; *Kurada and White, 1998*). However, in *Ras^V12*-glands hid expression gradually increased from 96 hr to 120 hr AED coinciding with the increase in nuclear disintegration (*Figure 6A*). Coexpression of Drs with *Ras^V12* significantly decreased hid expression at both time points, while Drs-knock-down in *Ras^V12*-glands increased hid expression even further already at 96 hr AED. Together with the reduction in hid expression upon JNK-inhibition in *jnk^DN;;Ras^V12*-glands, Drs emerges as a negative regulator of the apoptotic inducer hid by inhibiting JNK-signaling (*Figure 5—figure supplement 1B*).

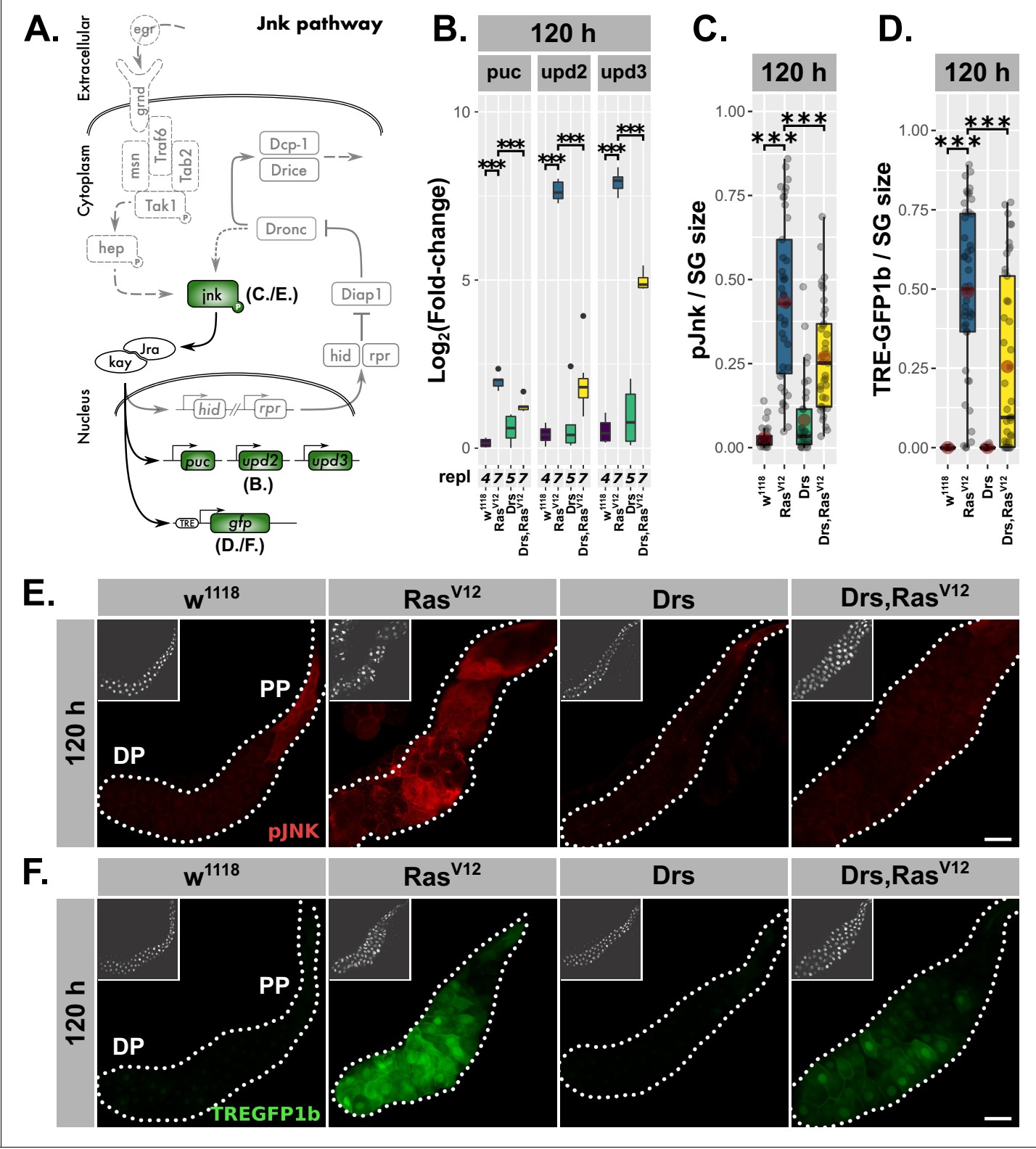

**Figure 5.** Drs overexpression inhibits JNK-activation. (**A**) Schematic representation of the JNK-pathway including read-outs (green) employed to track its activation. (**B**) qPCR results for canonical JNK-target genes (*log2*-transformed, fold-change over *Rpl32*) at 120 hr after egg deposition (AED). (**C**) Quantification of activated jnk by staining with phosphorylation-sensitive antibody normalized for the salivary gland (SG) size per gland at 120 hr AED. (**D**.) Quantification of TREGFP1b signal normalized for SG size per gland indicating JNK-dependent transcriptional activation at 120 hr AED. (**E**)

*Figure 5 continued on next page*

*Figure 5 continued*

Visualization of phosphorylated jnk in *Ras^V12*-glands with and without Drs-coexpression. (F) Distribution of TREGFP1b reporter signal in *Ras^V12*-glands in the presence or absence of coexpressed Drs. Insets: (E–F) DAPI. Scalebars: (E–F) 100 μm. Boxplots in (B–D): lower/upper hinges indicate 1st/3rd quartiles, whisker lengths equal 1.5*IQR, red circle and bar represent mean and median. Significance evaluated by Student's t-tests (***p<0.001).

The online version of this article includes the following figure supplement(s) for figure 5:

**Figure supplement 1.** JNK-signaling does not regulkate Drs expression.
**Figure supplement 2.** Drs levels determine JNK-activation.
**Figure supplement 3.** Overexpression of Drs is specific in its effect on JNK-signaling.

To evaluate the induction of PCD in *Ras^V12*-glands, we either inhibited JNK-signaling at the level of jnk activation or coexpressed the caspase-inhibitor p35 with *Ras^V12* and examined nuclear volume and integrity (*Figure 6B–D*). p35 inhibits Drice and thus blocks PCD at the level of effector caspase activation (*Hay et al., 1994*; *Meier et al., 2000*; *Hawkins et al., 2000*). While both interventions successfully blocked nuclear disintegration, the additional rounds of endoreplication as observed in *Ras^V12*-glands were not significantly suppressed (*Figure 6C–D*). Crucially, Drs-coexpression in *Ras^V12*-glands phenocopies the inhibition of JNK-signaling and effector caspases both in terms of restoring nuclear integrity as well as the persistence of excess endoreplications, in spite of a trend of overall nuclei size reduction similar to *p35;Ras^V12*-glands.

Last, to validate the inhibition of PCD by Drs, we monitored Dronc activity using the cleaved caspase 3 (CC3) antibody. In fact, the strongest Dronc activity occurred in cells that also displayed heavy disintegration of nuclei, confirming the relation between caspase activation and nuclear disintegration as part of PCD (*Figure 6—figure supplement 1B*). However, in *Drs,Ras^V12*-glands Dronc was significantly less activated compared to *Ras^V12*-glands, a phenotype which we showed to be independent of UAS-dilution effects (*Figure 6E–F*; *Figure 6—figure supplement 1A–D*; *Fan and Bergmann, 2010*). Given the strong reduction of Dronc activity upon knock-down of hid in *Ras^V12*-glands, the inhibitory effect of Drs on PCD is most likely to a large extent dependent on hid-inhibition (*Figure 6—figure supplement 1E–F*).

Taken together, we show that JNK-dependent induction of PCD is a consequence of the collapse in adaptation to *Ras^V12*-induced hypertrophy. Importantly, overexpression of Drs in *Ras^V12*-glands prevents PCD by blocking full JNK-activation. This is in contrast to recent observations where AMPs, including Drs, act pro-apoptically aiding in the elimination of tumor cells and limiting tumor size (*Araki et al., 2019*; *Parvy et al., 2019*).

## Drs inhibits the JNK-feedback loop

Various *Drosophila* models for tissue transformation have shed light on the functional separation of the JNK-pathway into an upstream kinase cascade leading to jnk-activation and a downstream feedback-loop converging again on jnk activation (*Figure 7A*; *Fogarty et al., 2016*; *Pérez et al., 2017*; *Shlevkov and Morata, 2012*; *Muzzopappa et al., 2017*).

Our results revealed a negative regulatory impact of Drs on JNK-signaling in *Ras^V12*-glands, but did not determine which part of the pathway is targeted by Drs. To answer this question, we uncoupled the feedback loop from the upstream kinase cascade by solely overexpressing hid (*Figure 7A*). Irrespective of the differing models for the signal propagation downstream of Dronc, this initiator caspase remains seminal for establishing the actual feedback with jnk (*Fogarty et al., 2016*; *Shlevkov and Morata, 2012*). Thus, we stained glands for activated Dronc and jnk after a pulse of hid-overexpression during the larval wandering stage (*Figure 7A*). As expected, *hid*-expressing glands showed highly elevated levels of activated Dronc, but also jnk, which indicates the presence of feedback activation. Moreover, both phenotypes were reversed upon coexpression of Drs during the hid-expression pulse (*Figure 7B–C,E*).

To clarify that Drs operates in a similar fashion during *Ras^V12*-induced hypertrophic growth, we stained *Ras^V12*-glands (*Figure 6E–F*) for activated jnk in the presence and upon knock-down of Dronc (*Figure 7D,F*; *Figure 7—figure supplement 1A–B*). In fact, the absence of a signal for activated jnk in *Dronc^RNAi;Ras^V12*- compared to strong *Ras^V12*-glands confirms the presence of a genuine feedback-regulation as part of JNK-signaling in SGs. Thus, Drs seems to inhibit JNK-signaling

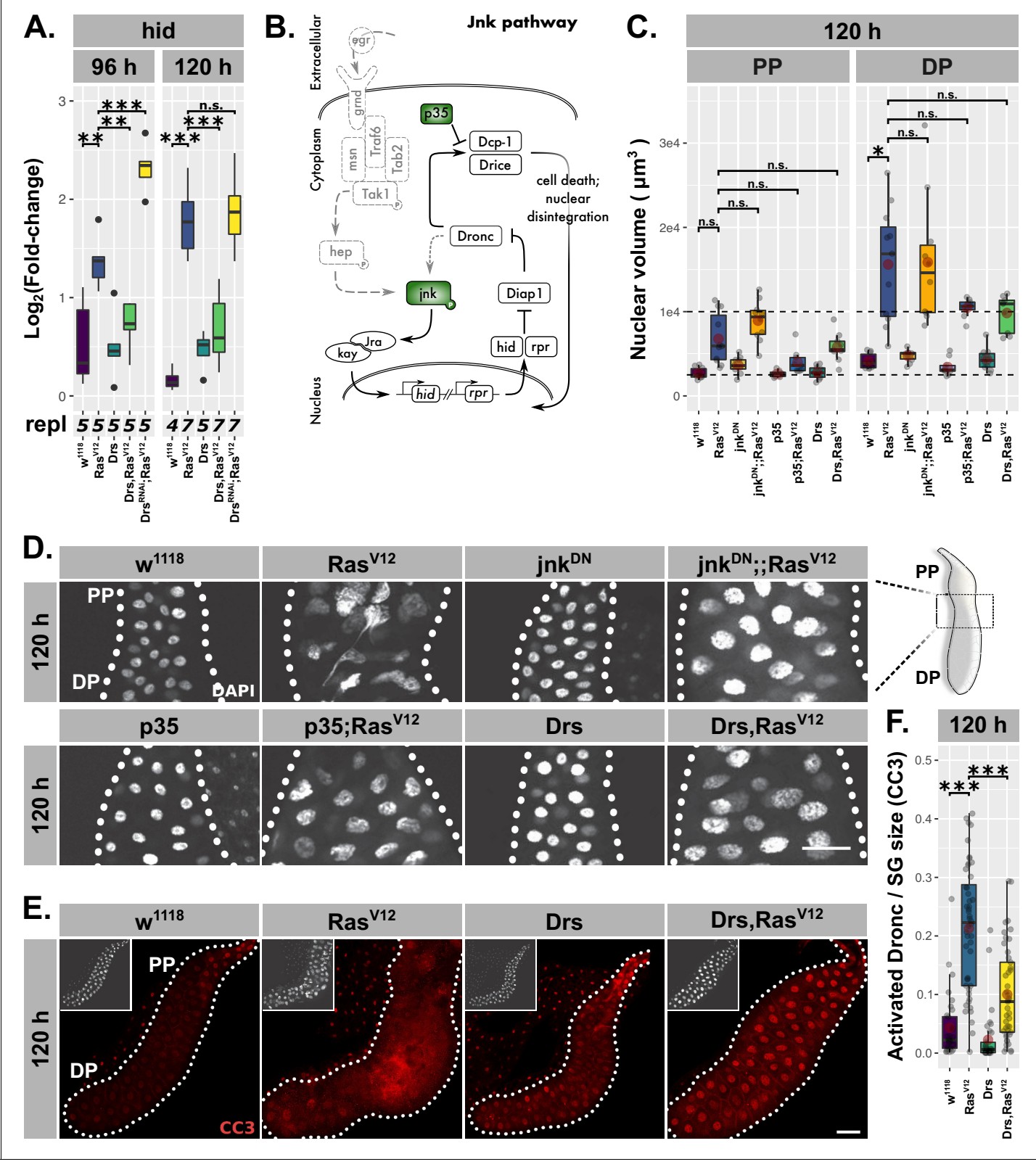

**Figure 6.** Drs inhibits programmed cell death. (**A**) Hid expression as measured by qPCR and plotted as *log2*-transformed values normalized to *Rpl32*-expression in *Ras^V12*-glands with in- (Drs) or decreased (Drs^RNAi) Drs expression. (**B**) Schematic representation of the JNK-pathway including used methods of interference (green). (**C**) Nuclear volumina derived from z-stacks of DAPI-stained salivary glands (SGs) and averaged per gland at 96 hr and 120 hr after egg deposition (AED). (**D**) DAPI-stained (white) SG nuclei to indicate nuclear size and disintegration. (**E–F**) *Ras^V12*-glands with increased Drs

*Figure 6 continued on next page*

*Figure 6 continued*
expression stained with anti-CC3-antibody to detect Dronc activation (red) and corresponding quantifications of detected signal normalized for SG size. Insets in (**E**) show DAPI and scalebars in (**D–E**) represent 100 µm. Lower/upper hinges of boxplots in (**A, C, F**) indicate 1st/3rd quartiles, whisker lengths equal 1.5*IQR, red circle and bar represent mean and median. Significance evaluated by Student's t-tests (***p<0.001).
The online version of this article includes the following figure supplement(s) for figure 6:

**Figure supplement 1.** Drs inhibits programmed cell death via hid and Dronc.

downstream of hid and upstream of Dronc and jnk, emphasizing an inhibition of the JNK-feedback loop rather than the initial kinase cascade.

## Discussion

### Tissue-autonomous vs. cellular immune response mediated via JNK-signaling

By overexpressing $Ras^{V12}$ in larval SGs, we made use of and overrode the gland's ability to adapt to growth signals. Activating constitutive Ras/MAPK-signaling allowed us to identify local immune and stress responses as part of a buffering mechanism to compensate the accumulation of stress and decipher natural limits of growth adaptation (*Hauling et al., 2014*). Remarkably, in this context, the local immune response inhibits the parallel stress response, an effect that to our knowledge has not been described before. Central to this inhibition is the AMP Drs, which directly impinges on the JNK-pathway and thereby subsequently also on inducing PCD, a function completely opposite to previous observations in other tissues (*Araki et al., 2019*; *Parvy et al., 2019*). These results also indicate an unprecedented role for an AMP as a signal transducer enabling tissue-autonomous crosstalk between immune and stress pathways. Dependent on the extent of JNK-inhibition by the local, Drs-dependent immune response and thus the integrated decision of both on the state of the tissue's homeostasis, the gland epithelium attracts hemocytes as part of a wider, cellular immune response. By virtue of inhibiting JNK-signaling, tissue-autonomous and cellular immune responses antagonize each other, a balance that determines continuous hypertrophic growth or its restriction (*Figure 8*). The latter has far reaching implications for therapeutic approaches that need to consider the adverse effects that stimulating immune responses might have on tissue growth after damage and under stress.

### Drs is expressed as part of a genuine tissue-autonomous immune response

The SGs of *Drosophila* larvae are an integral part of its gastrointestinal system and the lumen of the mature glands forms a continuum with the exterior. As such the glands are constantly exposed to microbial and pathogenic influences, which predestines them to act as a dedicated immunological barrier epithelium. However, raising larvae with $Ras^{V12}$-glands under strictly sterile conditions corroborated the authenticity of the immune response as tissue-autonomous and with high likelihood independent of exogenous or systemically distributed pathogenic stimuli. This is further emphasized by the dependency of Drs expression on the tissue-specific overexpression on $Ras^{V12}$ and tissue-autonomous manipulations (i.e. *Drs* or $jnk^{DN}$ overexpression) leading to the inhibition of JNK-activation and hemocyte recruitment. The activation of immune effectors in the absence of exogenous or endogenous pathogens is also consistent with the definition of sterile inflammation and aligns with finding in other models (*Shaukat et al., 2015*; *Kenmoku et al., 2017*; *Asri et al., 2019*). However, Drs is embedded into a wider spatio-temporally regulated activation of immune effectors and immune mechanisms providing immunocompetence to the PP until 120 hr AED and thus constituting a genuine immune response (*Figure 3C–D*; *Figure 3—figure supplement 1B–C*). Follow-up studies will determine the mode of their activation under physiological and sterile conditions and thus clarify their nature in the context of inflammation and immunity.

Apart from dl and Mef2 neither the homeobox transcription factor Caudal as in the adult glands nor any other of the canonical members belonging to the Toll- and imd-pathway are involved in Drs expression (*Hauling et al., 2014*; *Ferrandon et al., 1998*; *Ryu et al., 2004*). Under wild-type conditions, dl is expressed throughout the entire gland epithelium until 96 hr AED, but only remains

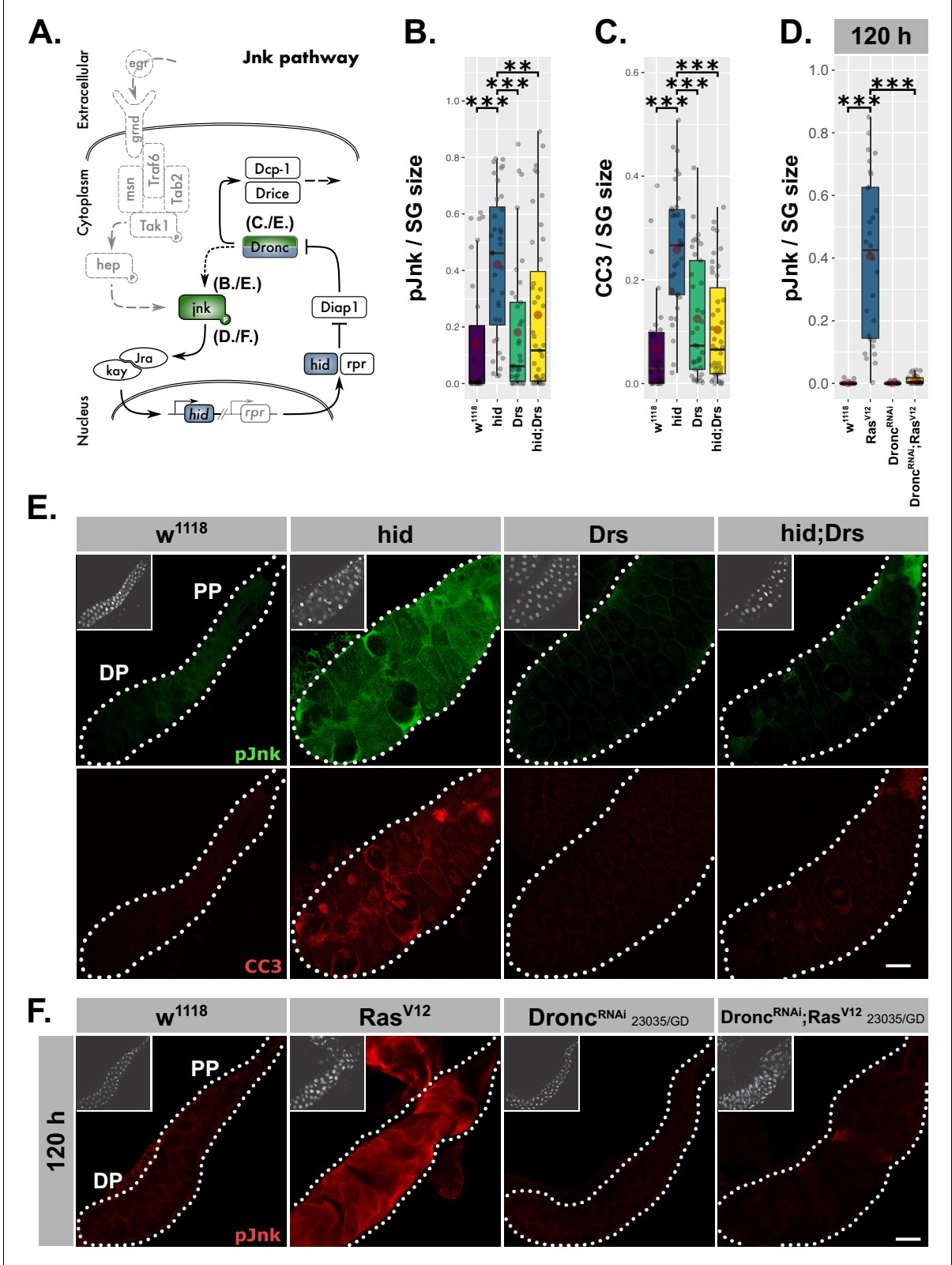

**Figure 7.** Drs inhibits the JNK-feedback loop. (**A**) Schematic representation of the JNK-feedback loop including levels of interference (blue) and used read-outs of its activity (green). *hid*-overexpressing glands with or without coexpressed Drs were quantified for (**B**) activated jnk via a phosphorylation sensitive antibody and (**C**) activated Dronc via the CC3-antibody. (**D**) Activated jnk (pJnk) was quantified and normalized to tissue size in *Ras^V12*-glands upon knockdown of Dronc. (**E**) Salivary glands solely overexpressing hid with or without Drs-coexpression were stained for activated jnk (green) and

*Figure 7 continued on next page*

*Figure 7 continued*

activated Dronc (red). (F) Activated jnk (red) was detected in *Ras^V12*-glands with or without Dronc-knock-down at 120 hr AED. Insets: (E–F) DAPI. Scalebars: (E–F) 100 µm. Boxplots in (B–D): lower/upper hinges indicate 1st/3rd quartiles, whisker lengths equal 1.5*IQR, red circle and bar represent mean and median. Significance evaluated by Student's t-tests (***p<0.001, **p<0.01).

The online version of this article includes the following figure supplement(s) for figure 7:

**Figure supplement 1.** Dronc-reduction blocks the JNK-feedback loop in *Ras^V12*-glands.

expressed in the PP at 120 hr AED (*Figure 2B*). This spatio-temporal expression pattern in turn determines to a large extent the expression of Drs in *Ras^V12*-glands essentially separating the gland at 120 hr AED into an immunocompetent duct-proximal and a stress-responsive duct-distal compartment (*Figure 2B*; *Figure 2—figure supplement 3A*; *Figure 8*). Due to the absence of Drs in the DP at 120 hr AED, stress-responsive JNK-signaling finally exceeds a critical threshold leading to elevated Mmp2- and hid-expression. This stimulates the onset of PCD and opening of the basal membrane as an important prerequisite for attachment of hemocytes that are subsequently recruited to the surface of the SGs (*Figure 4B,D*; *Figure 8*; *Hauling et al., 2014*). To the contrary, dl continues to induce Drs in the PP of *Ras^V12*-glands until 120 hr AED in line with the complete absence of hemocyte recruitment to this part of the gland (*Figure 1F*; *Figure 1—figure supplement 1G–H*). However, Drs becomes less dependent on dl between 96 hr and 120 hr AED. While its precise role in Drs-regulation especially in relation to dl remains to be elucidated, Mef2 might influence Drs-expression directly or indirectly parallel to ceasing dl-influence (*Figure 2—figure supplement 3B–C*).

## Drosomycin impinges on JNK feedback activation

In depth analysis of wound regeneration and tumor formation has shed light on the intricate architecture of the JNK-pathway and its signal propagation (*Santabárbara-Ruiz et al., 2015*; *Pinal et al., 2018*). This has led to the discovery of feedback- or self-sustenance-loops as part of JNK-signaling (*Fogarty et al., 2016*; *Pérez et al., 2017*; *Shlevkov and Morata, 2012*; *Muzzopappa et al., 2017*). Similarly, several lines of evidence validated the existence of a complex quantitatively and qualitatively regulated activation of the JNK-pathway in *Ras^V12*-glands. In fact, the strong reduction of jnk activation upon knock-down of the initiator caspase Dronc in the *Ras^V12*-background at 120 hr AED indicates that a feedforward-loop is also part of the propagation of JNK-activation in hypertrophic glands (*Figure 7D,F*; *Figure 7—figure supplement 1A–B*). Consistently, even sole hid overexpression in SGs led to Dronc as well as jnk activation emphasizing the presence of this feedback-regulation as part of the JNK-pathway further (*Figure 7B–C,E*).

In general, feedback loops serve as a predestined platform to integrate additional signals via crosstalk with other pathways to dynamically modulate the originally transmitted signal and thus either amplify or weaken the response (*Kholodenko, 2006*; *Antebi et al., 2017*). Here, we describe a novel mode of signal attenuation by the tissue-autonomous immune response that rather than eliminating an exogenous stimulus directly interferes with the signal propagation in the JNK-feedback loop. Fundamentally, this interaction is part of an emerging picture of crosstalk between immune and stress responses involved in organ growth and maintenance in *Drosophila* (*Wu et al., 2015*; *Liu et al., 2015*; *Parisi et al., 2014*; *Hauling et al., 2014*).

Drs regulates the signal propagation of the intracellular module of the JNK-pathway and throughout our experiments only directly Drs-expressing cells prevented *Ras^V12*-induced cell disintegration and PCD. However, it remains an outstanding question whether Drs functions exclusively cell-autonomously and intracellularly or whether secreted Drs operates in an autocrine manner too. Clonal analysis and rescue experiments will serve this purpose in the future. Further work on the effector mechanism of Drs will also elucidate further details about the components of the JNK-feedback loop in- or directly regulated by Drs. This will also contribute to mapping the manifold modes of immune-stress-crosstalk in *Drosophila* and find general patterns among them beyond a sole dependency on the specific context.

## Drs promotes hypertrophic growth and inhibits PCD

As our *Drs,Ras^V12*-experiments indicated, under conditions of continuous growth and therefore chronic stress induction, the ability to suppress the JNK-pathway in a Drs-dependent manner

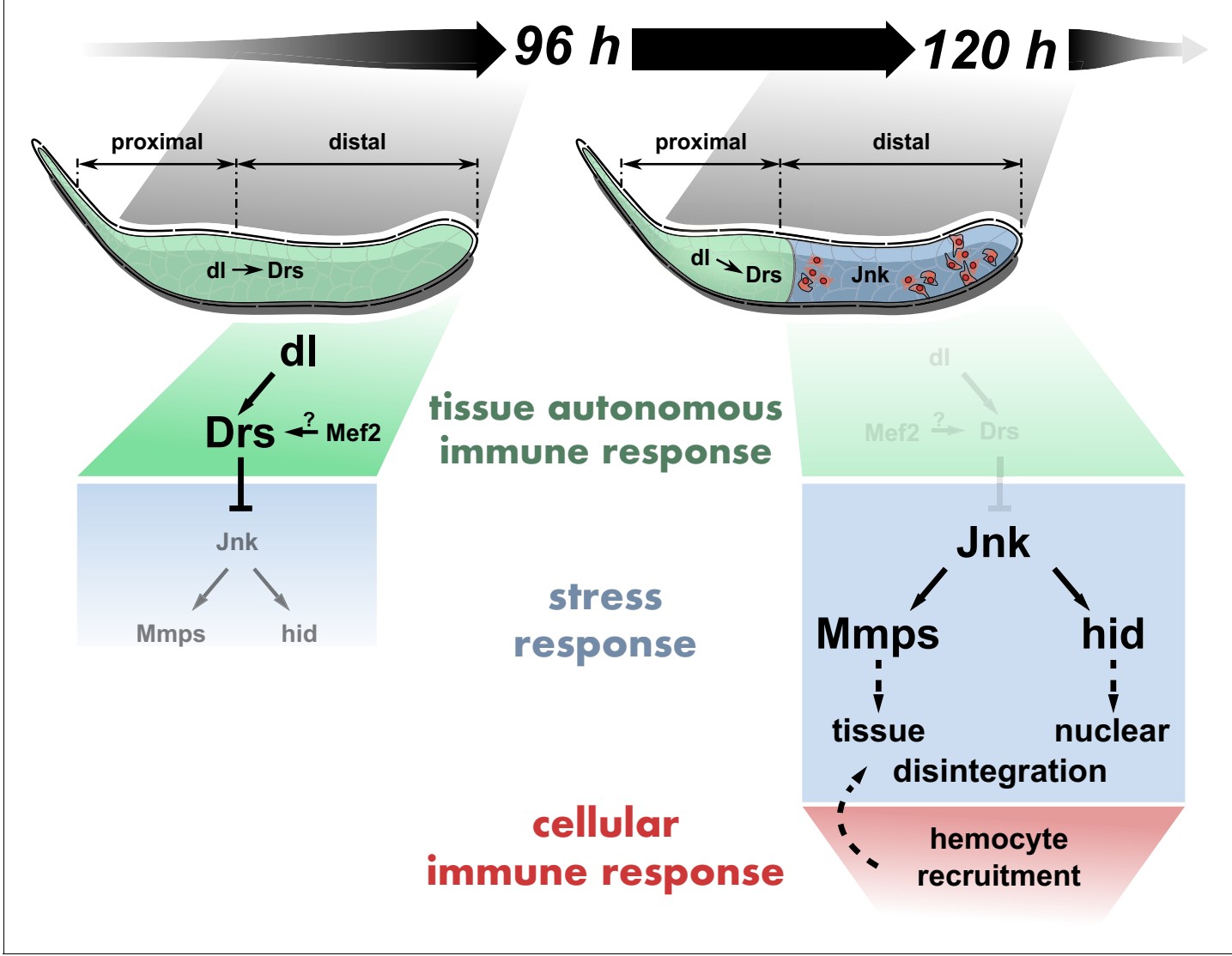

**Figure 8.** Tissue-autonomous antagonizes cellular immune response via JNK-inhibition in *Ras^V12*-glands . dl-mediated Drs expression inhibits JNK activation in the entire *Ras^V12*-gland until 96 hr after egg deposition (AED). At 120 hr AED, dl and thus Drs expression is reduced to the PP, derepressing full JNK-activation in the DP. Consequently, JNK-dependent expression of hid and Mmp2 participate in stimulating nuclear and tissue disintegration, which eventually leads to the interference with tissue growth and integrity by the cellular immune response.

supports continuous, hypertrophic growth of *Ras^V12*-glands and the survival of the tissue. Moreover, a prolongation of Drs-expression beyond its endogenous decrease inhibits the induction of PCD and the recognition of the cellular immune response and thus renders the hypertrophic gland unchallenged.

In fact, while Ras/MAPK-signaling in mitotic tissues crucially suppresses apoptosis by downregulating hid expression, this effect is revoked in hypertrophic *Ras^V12*-glands (*Bergmann et al., 1998*; *Kurada and White, 1998*). Although the reason for this difference remains to be elucidated, it is in fact Drs which operates as the inhibitor of apoptotic inducers in hypertrophic glands. This function is central to the suppression of PCD in *Ras^V12*-glands (*Figure 6A*).

This differs fundamentally from the pro-apoptotic function of AMPs, which was recently described for two tumor models. In both, disc (discs large, *dlg*) (*Parvy et al., 2019*) and leukemic tumors (mxc^mbn1) (*Araki et al., 2019*), AMPs were shown to target tumor cells and limit tumor size by inducing apoptosis. In addition to the differences between the tumorous tissues (i.e. proliferative discs

and lymph glands), the fact that Drs is induced locally in SGs and acts tissue-autonomously may explain its different activities.

## Hypertrophic SGs are a remarkable system to discover buffer mechanisms

Being incapable of cell proliferation, damaged postmitotic tissues cannot rely on regenerative cell plasticity like imaginal discs or stem cell-derived tissue regeneration as in the *Drosophila* adult mid-gut (*Ohlstein and Spradling, 2006*; *Herrera et al., 2013*; *Herrera and Morata, 2014*; *Schuster and Smith-Bolton, 2015*; *Ahmed-de-Prado and Baonza, 2018*). Instead, they need to cope with endogenous and exogenous influences via elaborate mechanisms to prevent or buffer detrimental consequences.

Here, we show that continued $Ras^{V12}$-expression in the larval SG overrides the dependency on nutritional cues and stimulates excess endoreplications that eventually have damaging effects on the tissue integrity by inducing elevated levels of stress. Uninterrupted induction of endoreplications has its natural limits in every system, even in SGs that are already polyploid. Eventually, continuous induction of more endocycles is challenged by nutritional restrictions in synthesizing more DNA, replication stress, spatial limitations in the gland nuclei and continuously more unsynchronized metabolic turnover. Hence, in spite of the anti-apoptotic function $Ras^{V12}$ conveys in mitotic tissues, unrestricted stimulation of excess endoreplications ultimately leads to cell death. Since this characterizes the final collapse of tissue homeostasis, it also allows to study the extent of buffering capacity conveyed by immune and stress-responsive signaling. While JNK-signaling has been shown to be seminal during development (i.e. tissue morphogenesis) (*Agnès et al., 1999*; *Zeitlinger and Bohmann, 1999*), its activation in $Ras^{V12}$-glands appears however rather likely to be due to its function in relaying stress signals, emphasizing the presence of stress as part of hypertrophic growth further.

Thus, hypertrophic $Ras^{V12}$-glands constitute an outstanding system to study the involvement of tissue-autonomous immune and stress-induced responses to buffer deviation from homeostasis.

## Immune surveillance theory

According to the immune surveillance theory, the immune system has evolved to reduce the risk of somatic cells accumulating cancerous mutations (*Burnet, 1957*; *Burnet, 1970*). In order to reduce the danger of cell-transformation, cells express or expose molecules upon recognition of stress or damage during transformation. These markers are sensed by the immune system, which in turn eradicates the potentially harmful cells (*Jung et al., 2012*; *Vantourout et al., 2014*; *Schmiedel and Mandelboim, 2018*). Given the absence of exogenous or endogenous pathogenic agents, the activation of immune effectors (i.e. Drs) in the here presented system is most likely a consequence of sensing Danger or Damage Associated Molecular Patterns during hypertrophic gland overgrowth (*Matzinger, 1994*; *Seong and Matzinger, 2004*). However, the immune surveillance theory remains controversial, since tumor-associated inflammation was also shown to promote rather than suppress tumor growth (*Balkwill and Mantovani, 2001*; *Mantovani et al., 2008*). Our model bridges the gap between these two opposing views, since the effects of the tissue-autonomous and cellular immune responses appear to be antagonistic regarding the regulation of JNK-activation and thus ultimately PCD. In fact, only the integration of the various stress and immune mechanisms in hypertrophic $Ras^{V12}$-glands allows a concerted decision to eradicate a putatively dangerous cell via inducing PCD or not. Given the evolutionary conservation from insects to mammals of signaling pathways that govern growth control (*Edgar, 2006*), it is likely that mechanisms to detect and counteract a loss in regulation of these pathways, such as stress and immune pathways, are similarly conserved between both phyla.

## Materials and methods

**Key resources table**

| Reagent type (species) or resource | Designation | Source or reference | Identifiers | Additional information |
|---|---|---|---|---|

*Continued on next page*

*Continued*

| Reagent type (species) or resource | Designation | Source or reference | Identifiers | Additional information |
|---|---|---|---|---|
| Genetic reagent (*D. melanogaster*) | UAS-dl^RNAi | Bloomington | 36650 | |
| Genetic reagent (*D. melanogaster*) | UAS-l(2)gl^RNAi | VDRC | 109604/KK | *Rives-Quinto et al., 2017* |
| Genetic reagent (*D. melanogaster*) | UAS-imd^RNAi | VDRC | 101834/KK | *Bosch et al., 2005* |
| Genetic reagent (*D. melanogaster*) | UAS-Fadd^RNAi | VDRC | 100333/KK | |
| Genetic reagent (*D. melanogaster*) | UAS-key^RNAi | VDRC | 100257/KK | |
| Genetic reagent (*D. melanogaster*) | UAS-Rel^RNAi | VDRC | 108469/KK | *Cammarata-Mouchtouris et al., 2020* |
| Genetic reagent (*D. melanogaster*) | UAS-spz^RNAi | VDRC | 105017/KK | *Panettieri et al., 2020* |
| Genetic reagent (*D. melanogaster*) | UAS-Tl^RNAi | VDRC | 1000788/KK | *Alpar et al., 2018* |
| Genetic reagent (*D. melanogaster*) | UAS-cad^RNAi | VDRC | 49562/KK | |
| Genetic reagent (*D. melanogaster*) | UAS-Stat92E^RNAi | VDRC | 106980/KK | *Recasens-Alvarez et al., 2017* |
| Genetic reagent (*D. melanogaster*) | UAS-Dronc^RNAi | VDRC | 100424/KK | *Kale et al., 2015* |
| Genetic reagent (*D. melanogaster*) | UAS-Dronc^RNAi | VDRC | 23035/GD | *Florentin and Arama, 2012* |
| Genetic reagent (*D. melanogaster*) | UAS-Myd88^RNAi | VDRC | 25402/GD | *Li et al., 2020* |
| Genetic reagent (*D. melanogaster*) | UAS-pll^RNAi | VDRC | 2889/GD | *Wu et al., 2015* |
| Genetic reagent (*D. melanogaster*) | UAS-Dif^RNAi | VDRC | 30578/GD | *Wu et al., 2015* |
| Genetic reagent (*D. melanogaster*) | UAS-Dif^RNAi | VDRC | 30579/GD | *Wu et al., 2015* |
| Genetic reagent (*D. melanogaster*) | UAS-Drs^RNAi | VDRC | 2703/GD | |
| Genetic reagent (*D. melanogaster*) | UAS-dfr^RNAi | S. Certel | | |
| Genetic reagent (*D. melanogaster*) | UAS-hid^RNAi | VDRC | 8269/GD | *Nagata et al., 2019* |
| Genetic reagent (*D. melanogaster*) | UAS-foxo^RNAi | VDRC | 107786/KK | *McLaughlin et al., 2019* |
| Genetic reagent (*D. melanogaster*) | UAS-grh^RNAi | VDRC | 33680/GD | |
| Genetic reagent (*D. melanogaster*) | UAS-Mef2^RNAi | Bloomington | 38247 | *Zhao et al., 2020* |
| Genetic reagent (*D. melanogaster*) | UAS-Nrf2^RNAi | VDRC | 101235/KK | *Brock et al., 2017* |
| Genetic reagent (*D. melanogaster*) | UAS-Nrf2^RNAi | VDRC | 108127/KK | *Brock et al., 2017* |
| Genetic reagent (*D. melanogaster*) | UAS-Sox14^RNAi | VDRC | 107146/KK | *Wang et al., 2020* |
| Genetic reagent (*D. melanogaster*) | DrsGFP | W.-J. Lee | | |

*Continued on next page*

*Continued*

| Reagent type (species) or resource | Designation | Source or reference | Identifiers | Additional information |
|---|---|---|---|---|
| Genetic reagent (*D. melanogaster*) | TRE-GFP1b | D. Bohmann | | |
| Genetic reagent (*D. melanogaster*) | UAS-Drs | B. Lemaitre | | |
| Genetic reagent (*D. melanogaster*) | UAS-CecA1 | B. Lemaitre | | |
| Genetic reagent (*D. melanogaster*) | dl$^{15}$ | Y. Engström | | |
| Genetic reagent (*D. melanogaster*) | Myd88$^{KG03447}$ | Y. Engström | | |
| Genetic reagent (*D. melanogaster*) | UAS-hid | M.Suzanne | | |
| Genetic reagent (*D. melanogaster*) | UAS-Mmp2$^{#4}$ | A. Page-McCaw | | |
| Genetic reagent (*D. melanogaster*) | UAS-Mmp1$^{APM1037}$ | A. Page-McCaw | | |
| Genetic reagent (*D. melanogaster*) | UAS-Mmp1$^{APM3099}$ | A. Page-McCaw | | |
| Genetic reagent (*D. melanogaster*) | UAS-Ras$^{V12}$ | Bloomington | 4847 | |
| Genetic reagent (*D. melanogaster*) | Bx$^{MS1096}$ | Bloomington | 8860 | |
| Genetic reagent (*D. melanogaster*) | UAS-jnk$^{DN}$ | Bloomington | 6409 | UAS-bsk$^{DN}$ |
| Genetic reagent (*D. melanogaster*) | tubP-Gal80$^{ts}$ | Bloomington | 7108 | |
| Genetic reagent (*D. melanogaster*) | Mmp2$^{k00604}$ | Bloomington | 10358 | |
| Genetic reagent (*D. melanogaster*) | 10xStat92E-GFP | Bloomington | 26197 | |
| Genetic reagent (*D. melanogaster*) | UAS-p35.H | Bloomington | 5072 | |
| Genetic reagent (*D. melanogaster*) | UAS-mCD8::mRFP | Bloomington | 27400 | |
| Genetic reagent (*D. melanogaster*) | vkg$^{G00454}$ | Flytrap | | Ref. 100; CollagenIV |
| Antibody | Anti-Hemese (mouse monoclonal) | István Andó | H2 | IF(1:5) |
| Antibody | Anti-pJNK (mouse monoclonal) | Cell Signaling Technology | Cat#:9255 | IF (1:250) |
| Antibody | Anti-cleaved caspase 3 (rabbit polyclonal) | Cell Signaling Technology | Cat#:9661 | IF(1:200) |
| Antibody | Anti-dorsal (mouse monoclonal) | DSHB | 7A4-39 | IF(1:50) |
| Antibody | Anti-mouse-IgG-Alexa546 (goat polyclonal) | ThemoFisher Scientific | Cat#: A-11030 | IF(1:500) |
| Antibody | Anti-rabbit-IgG-Alexa568 (goat polyclonal) | ThemoFisher Scientific | Cat#: A-11011 | IF(1:500) |
| Recombinant DNA reagent | Drs (cDNA clone) | DGRC | LP03851 | |
| Recombinant DNA reagent | CecA1 (cDNA clone) | DGRC | IP21250 | |

*Continued on next page*

*Continued*

| Reagent type (species) or resource | Designation | Source or reference | Identifiers | Additional information |
|---|---|---|---|---|
| Sequence-based reagent | oligo(dT)$_{16}$-primer | ThermoFisher Scientific | Cat#:8080128 | |
| Sequence-based reagent | Drs_F | This paper | qPCR primers | gaggagggacgctccagt |
| Sequence-based reagent | Drs_R | This paper | qPCR primers | ttagcatccttcgcaccag |
| Sequence-based reagent | AttD_F | This paper | qPCR primers | gtttatggagcggtcaacg |
| Sequence-based reagent | AttD_R | This paper | qPCR primers | tctggaagagattggcttgg |
| Sequence-based reagent | TIMP_F | This paper | qPCR primers | aacagagcgtcatggcttca |
| Sequence-based reagent | TIMP_R | This paper | qPCR primers | tcacaccaaaacaggtggca |
| Sequence-based reagent | Upd1_F | This paper | qPCR primers | cgggtgatcgcttcaatc |
| Sequence-based reagent | Upd1_R | This paper | qPCR primers | ctgcggtactcccgaaag |
| Sequence-based reagent | Upd2_F | This paper | qPCR primers | aagttcctgccgaacatgac |
| Sequence-based reagent | Upd2_R | This paper | qPCR primers | atccttgcggaacttgtactg |
| Sequence-based reagent | Upd3_F | This paper | qPCR primers | actgggagaacacctgcaat |
| Sequence-based reagent | Upd3_R | This paper | qPCR primers | gcccgtttggttctgtagat |
| Sequence-based reagent | Hid_F | This paper | qPCR primers | tctacgagtgggtcaggatgt |
| Sequence-based reagent | Hid_R | This paper | qPCR primers | gcggatactggaagatttgc |
| Sequence-based reagent | Rpr_F | This paper | qPCR primers | gatcaggcgactctgttgc |
| Sequence-based reagent | Rpr_R | This paper | qPCR primers | actgtgactcccgcaagc |
| Sequence-based reagent | Grim_F | This paper | qPCR primers | atcgatgaccatgtcggagt |
| Sequence-based reagent | Grim_R | This paper | qPCR primers | cgcagagcgtagcagaagat |
| Sequence-based reagent | MMP1_F | This paper | qPCR primers | gtttccaccaccacacagg |
| Sequence-based reagent | MMP1_R | This paper | qPCR primers | gcagaggcgggtagatagc |
| Sequence-based reagent | MMP2_F | This paper | qPCR primers | tttcgatgcggacgagac |
| Sequence-based reagent | MMP2_R | This paper | qPCR primers | gccacgttcagaaaattggt |
| Sequence-based reagent | PUC_F | This paper | qPCR primers | cgtcatcatcaacggcaat |
| Sequence-based reagent | PUC_R | This paper | qPCR primers | aggcggggtgtgtttctat |
| Sequence-based reagent | RPL32_F | This paper | qPCR primers | cggatcgatatgcta |

*Continued on next page*

*Continued*

| Reagent type (species) or resource | Designation | Source or reference | Identifiers | Additional information |
|---|---|---|---|---|
| Sequence-based reagent | RPL32_R | This paper | qPCR primers | cgacgcactctgttg |
| Peptide, recombinant protein | RNase-free DNase I | ThermoFisher Scientific | Cat#:EN0521 | |
| Peptide, recombinant protein | SuperscriptIII | ThermoFisher Scientific | Cat#:18080-093 | |
| Peptide, recombinant protein | Phalloidin-546 | Molecular probes | Cat#:A22283 | |
| Commercial assay or kit | KAPA SYBR FAST qPCR Master Mix (2x) kit | Kapa Biosystems; Sigma-Aldrich | KR0389, v9.13 | |
| Commercial assay or kit | RNAqueous-Micro Kit | ThermoFisher Scientific | Cat#:AM1931 | |
| Commercial assay or kit | RNAqueous Kit | ThermoFisher Scientific | Cat#:AM1912 | |
| Commercial assay or kit | Experion RNA StdSens Reagents and Supplies | Bio-Rad | Cat#:700-7154 | |
| Commercial assay or kit | Experion RNA StdSens Chips | Bio-Rad | Cat#:700-7153 | |
| Chemical compound, drug | Vancomycin | Sigma-Aldrich | Cat#:V2002 | |
| Chemical compound, drug | Metronidazole | Sigma-Aldrich | Cat#:M3761 | |
| Chemical compound, drug | Neomycin | Sigma-Aldrich | Cat#:N1876 | |
| Chemical compound, drug | Carbenicillin | Sigma-Aldrich | Cat#:C1389 | |
| Chemical compound, drug | Sodium Hypochlorite solution | Fisher Scientific | Cat#:10401841 | |
| Chemical compound, drug | DAPI | Sigma-Aldrich | D9542 | |
| Software, algorithm | ImageJ | Fiji contributors | v1.52n | https://imagej.nih.gov/ij/ |
| Software, algorithm | Zen software | Zeiss | Blue edition | |
| Software, algorithm | kallisto | *Bray et al., 2016* | v0.44.0 | https://pachterlab.github.io/kallisto/ |
| Software, algorithm | sleuth | *Pimentel et al., 2017* | v0.30.0 | https://pachterlab.github.io/sleuth/ |
| Software, algorithm | GOstats | *Falcon and Gentleman, 2007* | v2.48.0 | https://github.com/Bioconductor/GOstats/ |
| Software, algorithm | AnnotationDbi | *Pagès et al., 2020* | v1.44.0 | https://github.com/Bioconductor/AnnotationDbi/ |
| Software, algorithm | org.Dm.eg.db | *Carlson, 2019* | v3.7.0 | http://bioconductor.org/packages/org.Dm.eg.db/ |
| Software, algorithm | RNAseq_sg_analysis.Rmd | This paper | | https://github.com/robertkrautz/sg_analysis/; *Krautz, 2021*; copy archived at swh:1:rev:82c91040d3434b215c04cfce11cf73f70300e099 |
| Software, algorithm | RNAseq_motif_enrichment.Rmd | This paper | | https://github.com/robertkrautz/sg_analysis/ |
| Software, algorithm | scanner.ijm | This paper | | https://github.com/robertkrautz/sg_analysis/ |
| Software, algorithm | RcisTarget | *Aibar et al., 2017* | v1.6.0 | https://www.bioconductor.org/packages/release/bioc/html/RcisTarget.html |

*Continued*

| Reagent type (species) or resource | Designation | Source or reference | Identifiers | Additional information |
|---|---|---|---|---|
| Software, algorithm | biomaRt | *Durinck et al., 2009* | v2.42.1 | https://bioconductor.org/packages/release/bioc/html/biomaRt.html |
| Software, algorithm | Biostrings | *Pàges et al., 2020* | v2.54.0 | https://bioconductor.org/packages/release/bioc/html/Biostrings.html |
| Software, algorithm | PWMEnrich | *Stojnic and Diez, 2020* | v4.22.0 | https://www.bioconductor.org/packages/release/bioc/html/PWMEnrich.html |

## Fly husbandry and stocks

All crosses were reared on standard potatomash/molasses medium under tempered conditions (see 'Staging') in a 12 hr dark/12 hr light-cycle. Drs-GFP (W.-J. Lee), UAS-dfr$^{RNAi}$ (S.Certel), TRE-GFP1b (D. Bohmann), UAS-Drs and USA-CecA1 (B. Lemaitre), dl$^{15}$ and Myd88$^{KG03447}$ (Y. Engström), UAS-hid (M.Suzanne), UAS-Mmp2$^{#4}$, UAS-Mmp1$^{APM1037}$, and UAS-Mmp1$^{APM3099}$ (A. Page-McCaw) were kind gifts from the indicated donors. The following stocks were provided by the Bloomington *Drosophila* Stock Center: UAS-Ras$^{V12}$ (4847), Bx$^{MS1096}$ (8860), UAS-jnk$^{DN}$ (UAS-bsk$^{DN}$; 6409), tubP-Gal80$^{ts}$ (7108), Mmp2$^{k00604}$ (10358), 10xStat92E-GFP (26197), UAS-p35.H (5072), and UAS-mCD8::mRFP (27400). RNAi-lines were sourced from the Bloomington Stock Center and the Vienna *Drosophila* Resource Center and are detailed in *Supplementary file 3*. The generation of the CollagenIV flytrap line vkg$^{G00454}$ was described in *Morin et al., 2001*. Please, see *Supplementary file 1* for a complete list of experimental crosses.

## Staging

Virgins were collected for 3–7 days before setting crosses. Initially, crosses were kept on standard food without antibiotics for 48 hr at 25°C. Eggs were collected for 6 hr (Immunohistochemistry and qPCR) or for 2 hr (RNASeq) at 25°C. When necessary, precollections were performed for 2 hr at 25°C prior to the actual collection. Egg collections were incubated for 24 hr at 29°C or for hid experiments 48 hr at 18°C and replicates of 24 hatched 1st instar larvae were afterwards transferred to vials with 3 ml standard food supplemented with Neomycin (0.1 mg·ml$^{-1}$, Sigma-Aldrich, N1876), Vancomycin (0.1 mg·ml$^{-1}$, Sigma-Aldrich, V2002), Metronidazol (0.1 mg·ml$^{-1}$, Sigma-Aldrich, M3761) and Carbenicillin (0.1 mg·ml$^{-1}$, Sigma-Aldrich, C1389). After incubation at 29°C for another 72 hr (96 hr AED) or 96 hr (120 hr AED), 3$^{rd}$ instar larvae were prepared for dissection or pictures were taken of whole larvae with a Leica MZ FLIII Fluorescence Stereomicroscope. For hid experiments, transferred larvae were incubated for 197 hr at 18°C, shifted to 29°C for 12 hr and finally dissected. To exclude microbial contamination and maintain germ-free conditions, BxGal4;;DrsGFP/UAS-Ras$^{V12}$-eggs were dechorionated with a 50% Sodium Hypochlorite solution (Fisher Scientific, 10401841) immediately after collection and transferred to vials with apple-agar supplemented with Nipagin, Propionic acid and the same antibiotics as above. Larvae were then analyzed at 24 hr and 48 hr AED.

## Drs reporter assay

Replicates of *Drosophila* larvae (n = 24 larvae·replicate$^{-1}$; N > 6 replicates) at 96 hr AED were screened for Drs-GFP reporter signals. Three phenotypes were distinguished: (1.) 'Full' SG pattern includes GFP signal across the entire PP and GFP$^{+}$-cells with reduced intensity in the DP. (2.) 'Partial' SG is GFP$^{+}$ throughout the PP, but less pronounced in DP with reduced signal intensity and fewer GFP$^{+}$-cells. (3.) 'None' phenotype lacks signal throughout the entire gland. Distribution of phenotypes were scored per replicate and significance calculated for the 'None'-phenotype via Dunn's test after performing the Kruskal-Wallis rank sum test.

## qPCR

Total RNA of dissected SGs was isolated with the RNAqueous-Micro Kit (ThermoFisher Scientific, AM1931) and from whole larvae with the RNAqueous Kit (ThermoFisher Scientific, AM1912) according to the instructors manual. Residual genomic DNA was digested with RNase-free DNaseI (ThermoFisher Scientific, EN0521) and cDNA reverse transcribed with SuperscriptIII (ThermoFisher Scientific, 18080–093) while using oligo(dT)$_{16}$-primer (ThermoFisher Scientific, 8080128). Quality of

all prepared totalRNA-extractions was evaluated on a 5 mM Guanidinium Thiocyanate-agarose gel, for optimization purposes on a BioRad Experion system (RNA StdSens Assay, 7007153, 7007154) and totalRNA for sequencing was run on a 2100 Bioanalyzer Instrument (Agilent Technologies, 5067–1511). qPCR reactions were set as technical triplicates with KAPA SYBR FAST qPCR Master Mix (KR0389, v9.13) including 200 nM final concentration of forward and reverse primers and run on a Rotor-Gene Q 2plex HRM machine (9001550). See *supplementary file 2* for list of all used qPCR primers. The number of biological replicates acquired and assayed in the experiment represented in the individual figures are denoted underneath the respective graphs.

## RNASeq library preparation and analysis

To avoid variability and thus confounding influences among the various RNASeq sample groups, we controlled rigorously for age of female parents, larval density to avoid larval crowding, age differences as well as developmental age itself and bacterial influences by using axenic culture conditions. For each genotype, three biological replicates were dissected with 30 pairs of SGs for the three groups including whole glands and 60 pairs of PPs for the corresponding group to acquire 5 μg of total RNA. No power or sample size calculations were performed. Replicate numbers were determined by the available resources and funds.

Poly(A)-containing mRNA molecules from totalRNA-samples were purified with oligo(dT)-magnetic beads, subsequently fragmented and cDNA synthesized with random primers using the TruSeq RNA Sample Preparation Kit v2. Adapter ligation and PCR-amplification precede cluster formation with a cBot cluster generation system. All samples were sequenced on a HiSeq 2500 Illumina Genome Sequencer as PE50. Reads were pseudoaligned with kallisto (v0.44.0) to a transcriptome index derived from all *Drosophila* transcript sequences of the dmel release r6.19. Subsequent analysis of transcript abundances was performed in R with sleuth (v0.30.0) including principal component analysis for dimensionality reduction, statistical and differential expression analysis based on the beta statistic derived from the wald test. Enriched gene ontology terms were identified by calculating hypergeometric p-values via the GOstats (v2.48.0) R package. Gene IDs were converted via the AnnotationDbi (v1.44.0) package and the reference provided with the org.Dm.eg.db (v3.7.0) database and geneset intersections visualized as UpSetR plot (v1.3.3). Please, see the accompanied R markdown (i.e. 'RNAseq_sg_analysis.Rmd') deposited on GitHub for details (https://github.com/robertkrautz/sg_analysis). The data is available under the accession number GSE138936 on the NCBI Gene Expression Omnibus.

## Detection of enriched motifs and Mef2 visualization

Significantly, differentially up- (i.e. q < 0.001 and b > 1) or downregulated (i.e. q < 0.001 and b < −1) genesets were determined for all three sample populations by subsetting q- and b-values as determined by sleuth with the latter indicating fold-change over expression in BxGal4>w[1118]-control crosses. The RcisTarget R package (v1.6.0) together with the accompanied file including all dm6 motif rankings (i.e. 'dm6-5kb-upstream-full-tx-11species.mc8nr.feather') were used for motif enrichment analysis in the various genesets and adding annotations. Enriched motifs were manually screened for functionally related motif groups (e.g. 'NFkB' with dl-, Dif-, Rel-relaetd motifs) and motifs plotted according to their affiliation to these groups.

The sequence of the 'dm6' genome assembly was acquired as fasta-file from the Ensembl database and nucleotide sequence frequencies across the genome calculated via customized functions including all major chromosomes. In parallel, promoter sequences for all 'dm6' genes were collected via the biomaRt package (v2.42.1) and converted into a DNAStringSet with the Biostrings package (v2.54.0). A curated list of all 'dm6' transcription factor motifs as position probability matrices was obtained from the Cis-BP database and converted to position weight matrices (PWMs) with the help of the PWMEnrich R package (v4.22.0) by using the above cacluated nucleotide sequence frequencies. Based on these PWMs and all promoter sequences, background scores for all 'dm6' transcription factor motifs across all promoters of the 'dm6' genome were quantified. Sequences for the wider Drs-locus and the shared Dif-/dl-loci environments were obtained via biomaRt and the enrichment of the Mef2-motif 'M08214_2.00_Mef2' compared to all background scores was calculated via PWMEnrich. Mef2 motif enrichment scores were plotted as a function of the location of the detected motif in the screened loci (i.e. Dif/dl; Drs) with the help of the novel, devised function

geom_bar_rastr(), which servers as an extension to the ggrastr R package. For details please follow the entire computational workflow as provided in 'RNAseq_motif_enrichment.Rmd' deposited on GitHub (https://github.com/robertkrautz/sg_analysis).

## Immunohistochemistry

PPs of staged larvae were inverted in PBS, unnecessary organs removed and samples fixed in 4% paraformaldehyde for 20 min. Subsequently, samples were washed three times for each 10 min in PBS or PBST (1% TritonX-100). Blocking was performed with 0.1% BSA in PBS (H2) or 5% BSA in PBST (anti-CC3, anti-pJNK and anti-dl). Samples were then incubated in primary antibodies dissolved in blocking buffer for 12 hr at 4°C or 1 hr at room temperature (RT). Primary antibodies comprised rabbit anti-cleaved caspase 3 (1:200, Cell Signalling Technology, 9661), mouse anti-pJNK (1:250, Cell Signalling Technology, 9255), mouse anti-dl (1:50, DSHB, 7A4), and mouse anti-Hemese (1:5; gift from István Andó). After washing as prior to blocking, secondary antibodies were applied together with DAPI (1:500; Sigma-Aldrich; D9542) and when necessary Alexa Fluor 546 Phalloidin (1:500, ThermoFisher Sceintific, A22283) in blocking buffer for 1 hr at RT. As secondary antibodies goat anti-Mouse-IgG-Alexa546 (H+L; 1:500; ThermoFisher Scientific; A-11030) and goat anti-Rabbit-IgG-Alexa568 (H+L; 1:500; ThermoFisher Scientific; A-11011) were employed. Final washing as prior to blocking preceded dissection of the samples in PBS and separation of SGs. Tissues were mounted in Fluoromount-G (SouthernBiotech) and analyzed with a Zeiss LSM780 confocal microscope. Images were extracted with Zen software (Blue edition) for further processing either in Adobe Photoshop CS5 Extended (v12.0.4 × 64) or Inkscape (v0.92).

## ROS-staining

H2DCFDA (D399, Thermo Fisher Scientific) was reconstituted in DMSO just prior to the experiment. This stock solution was further diluted to 10 μM in PBS. Freshly dissected glands were incubated in the 10 μM- working solution for 30 min in the dark. Three 5-min wash steps in PBS ensured removal of excess H2DCFDA-dye. Glands were mounted with PBS and confocal pictures obtained with the Zeiss LSM780 microscope system.

## Hemocyte- / pJNK- / CC3- quantification and size measurement

Pictures of stained glands were taken with a Zeiss Axioplan two microscope equipped with an ACH-ROPLAN ×4 lens and a Hamamatsu ORCA-ER camera (C4742-95). Images were extracted with Axio-Vision software (v40V 4.8.2.0) and analyzed with ImageJ. Cumulated area of hemocyte attachment, pJNK or CC3 fluorescence per SG or per SG part was filtered in the Red-channel and gland size determined by outlining glands, PPs or DPs with the 'Polygon selection'-tool. The distribution of hemocytes in *Drosophila* can be approximated by a natural logarithm, which required transformation of hemocyte attachment- and SG-areas before calculating ratios (*Sorrentino, 2010*). Normality across all samples of a particular genotype and where necessary separated by gland part (i.e. proximal or distal) or time (i.e. 96 hr and 120 hr AED) was evaluated with the Shapiro-Wilk-test and by bootstrapping via the fitdistrplus R package (v1.0.11). Significant differences between experimental groups were determined for pairwise comparisons via the Student's t-test after validating un-/equal variance via the Bartlett's test. Data and statistical analysis was performed in R. No power or sample size calculations were performed. All experiments were performed at least twice, while preparing and acquiring meaurements for all specimens and replicates of the respective experimental set-up. The amount of replicates was determined as a trade-off between reproducibility (maximum number) and feasibility of the experimental procedure to ensure same treatment and timing for all specimens. No measurements were categorized as outliers or excluded. Effect sizes were calculated by normalizing the effect and the reference samples by the mean of the reference samples, which anchors the reference mean at '1' and reports all deviations from it in procent.

## Nuclei volume

Z-stacks of entire SGs were captured with a Zeiss LSM780 confocal microscope for DAPI signal. Obtained stacks were further processed in ImageJ via the '3D Objects Counter'-plugin (v2.0.1.). Proximal and distal compartments were defined with the 'Polygon selection'-tool. Transfer of the region of interest to all z-stack slices, signal thresholding, object identification and volume

determination were integrated in a macro workflow. Data were plotted as average nucleus volume for all nuclei of individual glands or gland parts. Representative sections of individual z-stack slices showing the transition between proximal and distal compartments were cropped and added for illustration. Statistics were performed similar to the analysis of Hemocyte-, pJNK- and CC3-quantifications.

## Drs-dl-correlation

BxGal4;;DrsGFP>Ras$^{V12}$-glands were stained for dl (mouse anti-dl-Ab; 1:50, DSHB, 7A4) and genomic DNA (DAPI), whereafter z-stacks were obtained on a Zeiss LSM780 confocal microscope. Proximal and distal gland compartments were delineated as regions of interest (ROI) in ImageJ with the 'Polygon selection'-tool and nuclei therein defined via their DAPI-signal and the '3D Objects counter'-ImageJ plugin. Transfer of the 3D-nulei outlines across the dl- and Drs-channels via the ImageJ 'RoiManager 3D' (v3.96) allowed quantifying an accumulated intensity of the corresponding signals per nucleus. Data analysis was conducted in R including calculation of linear regression and Pearson coefficients for correlations between nuclei-wide Drs- and dl-signal for individual gland compartments and visualization of all nuclei data points after log$_{10}$-normalization for all five screened samples.

## Driver expression strength

DrsGFP-reporter or Phalloidin-stained, GFP-glands were captured with the 'Zeiss Axioplan two microscope / ACHROPLAN 4x lens / Hamamatsu ORCA-ER camera (C4742-95) / AxioVision software (v40V 4.8.2.0)'-system. ROIs were added manually to the imaged glands to outline their position in ImageJ with the 'Polygon selections'-tool. Feret diameter and angle were measured to delineate the longitudinal axis of each gland. For each position along the longitudinal axis, the corresponding rectangular line region intersecting with the glands ROI is determined and the encapsulated average fluorescence intensity measured. This workflow is implemented as ImageJ-macro (i.e. 'scanner.ijm') for screening of entire picture batches and available on GitHub (https://github.com/robertkrautz/sg_analysis). The resulting data was further analyzed in R to bin and align measurements along the longitudinal axis. Measurements were further averaged and normalized per bin across all screened glands. Given the statistically limited amount of observations (i.e. 200–350 bins), a loess function was used for smoothing the mean during visualization as implemented in geom_smooth() (i.e. span = 0.05).

## Probe synthesis and in situ hybdridization

cDNAs for the Drs (LP03851) and CecA1 (IP21250) loci were obtained from DGRC and plasmids transformed into DH5α according to the supplier's instructions. Extracted plasmids were linearized (EcoRI 20 U/µl, 10x EcoRI buffer, BSA 20 mg/ml in ddH2O), purified with phenol:chloroform:isoamyl alcohol (24:24:1/125:24:1) and precipitated (3M sodium acetate, ice-cold 100% Ethanol) for 30 min. In vitro transcription (5x transcription buffer, 1 µg template DNA, 10x Digoxigenin-11-UTP rNTP; Roche; 11277073910, 20U Ribolock; Fermentes; EO0381, T7 polymerase; ThermoFisher Scientific; EP0111) was performed for 3 hr at 37℃. The remaining DNA template was digested with DNase I for 15 min at 37℃ and digestion terminated with ammonium acetate (DEPC-H$_2$O:7.8M NH$_4$Ac:100% Ethanol – 1:0.5:3). The transcribed RNA was precipitated, purified and diluted in hybridization buffer (6.57 mg/ml Torula RNA; Sigma-Aldrich; R6625, 65% deionized formamide, 6.5x Saline sodium citrate [SSC], 65 µg/ml heparin, 0.1% Tween-20).

SGs were dissected in PBS, fixed in paraformaldehyde (4% PFA in PBSTw: 0.1% Tween-20 in PBS) for 40 min at room temperature (RT), washed four times in PBSTw for 5 min and transferred to Methanol. Permeabilization was performed for 1 hr in Ethanol, followed by washing once in Methanol for 5 min and twice in PBSTw for each 5 min. The samples were treated with Proteinase K (30 µg/ml; ThermoFisher Scientific; EO0491) for 6 min, washed twice with glycine (2 mg/ml) for 1 min, fixed for 20 min in 4% PFA and washed four times in PBSTw for each 5 min.

Prior to hybridization, glands were pre-treated with hybridization buffer (HYB: 5 mg/ml Torula RNA, 50% deionized formamide, 5x SSC, 50 µg/ml heparin, 0.1% Tween-20) for 1 hr before incubation with either CecA1- or Drs-probe overnight, both at 60℃ (in HYB with 3% Dextran). To avoid unspecific binding, samples were stringently washed at 60℃ once with HYB-wash solution (50%

formamide, 2x SSC, and 0.1% Tween-20) for 30 min, twice with 2x SSCTw, and trice with 0.2 x SSCTw for 20 min. At RT, samples were washed once with PBSTw for 5 min, blocked with sheep serum (8% in PBSTw; Sigma-Aldrich; S2263) for 1 hr and incubated with alkaline phosphatase coupled to anti-digoxigenin-antibody (1:4000 in PBSTw with 8% sheep serum; Roche; 1093274) at 4°C. Washing proceeded four times with PBSTw for each 15 min, twice with TNTw buffer (Tris-NaCl-Tween – 0.1M:0.1M:0.1%; pH 8) for each 15 min and twice with alkaline phosphatase buffer (100 mM Tris-HCL pH 9.5, 100 mM NaCl, 50 mM $MgCL_2$, 0.1% Tween-20) for each 5 min. Staining was perfomed with BCIP (5-bromo-4-chloro-3-indolyl-phosphate 4-toluidine salt: 0.175 mg/ml; Roche; 11383221001), NBT (4-nitro-blue-tetrazolium chloride: 0.3375 mg/ml; Roche; 11383213001) and levamisole (0.001 M) in alkaline phosphatase buffer. Stained samples were washed with alkaline phosphatase buffer for 5 min, TNTw for 15 min and PBSTw for 15 min. Washed samples were mounted in Fluoromount G (SouthernBiotech) and capturesd with Leica Firecam (v3.4.1) using a Leica DFC300x FX digital color camera coupled to a Leica MZ16 microscope. All experiments were performed at least twice including all specimens dissected from the entire set of replicates. The amount of replicates was determined as a trade-off between reproducibility and the amount of samples that could be processed with specific equipment of limited supply (i.e. sieves for sample transfer between buffers).

## Acknowledgements

We thank S Höglund and the Imaging facility at Stockholm University for microscopy support, the Bloomington *Drosophila* Stock Center and the Vienna *Drosophila* Resource Center for fly stocks, the *Drosophila* Genomics Resource Center for the Drs and CecA1 cDNAs, the Developmental Studies Hybridoma Bank for the anti-dl antibody and D Bohmann, S Certel, Y Engström, W-J Lee, B Lemaitre as well as M Suzanne for providing us with flies. We are especially grateful to R Karlsson, I Söll and G Hauptmann for decisive feedback on experimental procedures and J van den Ameele and the three reviewers for their critical review of the manuscript. This work was supported by the Swedish Research Council (VR-2010–5988 and VR 2016–04077) and the Swedish Cancer Foundation (CAN 2010/553 and CAN 2013/546).

## Additional information

### Funding

| Funder | Grant reference number | Author |
| --- | --- | --- |
| Vetenskapsrådet | VR-2010-5988 | Ulrich Theopold |
| Vetenskapsrådet | VR 2016-04077 | Ulrich Theopold |
| Swedish Cancer Foundation | CAN 2010/553 | Ulrich Theopold |
| Swedish Cancer Foundation | CAN 2013/546 | Ulrich Theopold |

The funders had no role in study design, data collection and interpretation, or the decision to submit the work for publication.

### Author contributions

Robert Krautz, Conceptualization, Data curation, Software, Formal analysis, Supervision, Validation, Investigation, Visualization, Methodology, Writing - original draft, Project administration, Writing - review and editing; Dilan Khalili, Data curation, Validation, Investigation, Visualization, Writing - review and editing; Ulrich Theopold, Conceptualization, Supervision, Funding acquisition, Project administration, Writing - review and editing

### Author ORCIDs

Robert Krautz (iD) https://orcid.org/0000-0003-0457-1348
Dilan Khalili (iD) http://orcid.org/0000-0002-9785-9641
Ulrich Theopold (iD) https://orcid.org/0000-0002-1009-8254

**Decision letter and Author response**
Decision letter https://doi.org/10.7554/eLife.64919.sa1
Author response https://doi.org/10.7554/eLife.64919.sa2

## Additional files

### Supplementary files
• Supplementary file 1. Complete list of all crosses associated with the experimental results in the indicated figures.

• Supplementary file 2. Complete list of all sequences corresponding to forward and reverse primers used for qPCR.

• Supplementary file 3. Overview of all RNAi-lines used in the DrsGFP-reporter assay, including references indicating prior use and evidence for active inhibition of the outlined target of the respective RNAi-line.

• Transparent reporting form

### Data availability
All sequencing data has been deposited at NCBI GEO under the record GSE138936.

The following dataset was generated:

| Author(s) | Year | Dataset title | Dataset URL | Database and Identifier |
|---|---|---|---|---|
| Krautz RK, Khalili DK, Theopold T | 2019 | Transcriptomes of whole hypertrophic *Drosophila* salivary glands and separated proximal gland parts | https://www.ncbi.nlm.nih.gov/geo/query/acc.cgi?acc=GSE138936 | NCBI Gene Expression Omnibus, GSE138936 |

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
