## [Decision Letter]

**Acceptance summary:**

Krautz et al. demonstrate that a tissue-autonomous innate immune response regulates hypertrophic growth using a *Drosophila* salivary gland model. A key finding is that the antimicrobial peptide Drosomycin inhibits the Jun-Kinase apoptotic feedback loop, permitting continued hypertrophic growth.

**Decision letter after peer review:**

[Editors’ note: the authors submitted for reconsideration following the decision after peer review. What follows is the decision letter after the first round of review.]

Thank you for submitting your work entitled "Tissue-autonomous immune response regulates stress signalling during hypertrophy" for consideration by *eLife*. Your article has been reviewed by three peer reviewers, and the evaluation has been overseen by a Reviewing Editor and a Senior Editor. The reviewers have opted to remain anonymous.

Our decision has been reached after consultation between the reviewers. Based on these discussions and the individual reviews below, we regret to inform you that the decision was to reject the manuscript while allowing re-submission. In this way, you are more than two months to submit a revised version that address all the point of the reviews. Importantly, you submission will be considered as a new submission but is likely to reviewed by the same reviewers. At this stage, the reviewers that important control are lacking to ascertain the solidity of the claim. If you feel that you will not be able to address the reviewer's comments, you can decide to transfer the manuscript to another journal.

Reviewer #1:

This work proposes that tissue-autonomous Drosomycin (Drs) negatively regulates *JNK* signalling in a hypertrophic salivary gland (SG) model. The flow of experiment ideas is thorough, but lacking critical control treatments. There appears to be a spatial bias in the BxGal4 expression model used. Additionally, there is no demonstration that this is Drs-specific, as the authors never test for the effect of Gal4 dilution by involving additional UAS constructs such as other AMPs or a control like UAS-GFP. As the paper relies entirely on a BxGal4>UAS model, alternate Gal4 and UAS control treatments are essential to avoid spatial bias and the effect of multiple UAS diluting RasV12 production and consequent hypertrophy.

List of substantive concerns:

1) The authors use a BxGal4 model to drive SG expression throughout this manuscript. The assumption seems to be that BxGal4 is a pan-SG driver, and that differences in the proximal and distal parts of the SG arise due to differences in Drs expression in these regions. However it is revealed in Figure 2B that BxGal4 does not efficiently drive dorsal-RNAi in the proximal part (PP) of the SG, as this tissue retains abundant dorsal expression. In Figure 4A there is a weakening of Drs signal in situ in the BxGal4>Drs at the PP, indicating BxGal4 is not efficiently driving Drs in the PP. Finally, the PCA analysis of RNAseq data shows totally independent clustering of the PP transcriptome from whole SGs; no BxGal4>*w1118* PP control is given for comparison.

Thus, throughout the manuscript, the PP is very likely a region with significantly reduced RasV12 expression, readily explaining most differences between the sick and apoptotic DP and the (relatively) healthy PP. For instance, this would readily explain differences in hemocyte attachment between the DP and PP. This may also imply a Drs response is seen in the PP solely because this relatively healthy tissue remains immune competent for longer than the DP.

Finally, Drs expression in the SG increases between the L3 and prepupa stages of development (FlyAtlas). It is unclear how BxGal4>RasV12 might affect SG developmental patterning, and indeed the model relies on rapid overgrowth of the tissue. Moreover there is an expected Drs response in immune-responsive tissues to sterile injury or stress. It is likely that SG remodelling during RasV12-induced stress leads to corresponding programmed cell death and similarly induction of NF-κB. This interpretation is congruous with the findings of Araki et al. (2018) and Parvy et al. (2019) that suggest NF-κB effectors cooperate in clearing tumorous tissue and responds to cell death signals.

2) Throughout the manuscript, the authors combine additional UAS constructs without controlling for the dilution of their Gal4 drive. This affects almost every figure in the latter half of the manuscript, and the effect of this dilution is apparent in Figure S5C, where Drs levels in BxGal4>Drs alone are 1-2 fold higher than BxGal4>Drs,RasV12 double drive. If RasV12 is diluted in combined UAS lines, this would promote a delayed phenotype, resulting in an apparent rescue effect if you simply compare equal time points. But it would be expected that continued monitoring to 144h would reveal the same apoptotic effects as the lone BxGal4>RasV12 tissue at 120h, similar to how BxGal4>RasV12 at 96h displays less apoptosis than 120h.

There is one argument that can be made in favour of the dilution not mattering: Drs-RNAi has a mild but opposite effect at 96h. However it should be noted that RNAi vs. protein overexpression leads to significantly different metabolic burdens on the cell, and thus RNAi is not a good control for this dilution effect. The authors should use a control overexpression construct such as UAS-GFP or UAS-RFP throughout.

3) There is no logic presented within the paper that this effect should be Drs-specific. Many other AMPs and immune effectors were similarly upregulated in their transcriptome (e.g. AttD, Def). If AttD and Def are similarly induced in this model, it is assumed their expression is similarly regulated, and that overexpression of AttD or Def could similarly rescue RasV12 SGs, and RNAi of AttD or Def could similarly exacerbate RasV12 SGs. I would recommend that, alongside a UAS-GFP control, the authors also test at least some subset of NF-κB effectors alongside their Drs model to elucidate if this is specific to Drs or a general effect. Given findings in other papers on AMP-tumor interactions where overexpressing Drs, Dpt, or Def all had similar effect (Araki et al., 2019), it seems likely that many AMPs should result in the same phenotype, implying there is no molecular pathway where specifically Drs inhibits *JNK*.

Reviewer #2:

Krautz et al. presents an interesting finding on regulation of stress signaling during hypertrophy via tissue-autonomous immune signaling. Data shows that Drosomycin, downstream of the NFkB factor Dorsal, interferes with overgrowth of the salivary gland caused by RasV12 expression. The authors investigate the underlying mechanism of this finding using different approaches, presenting copious data which are generally supportive their model.

However, in some cases the findings lack direct demonstration of key aspects of their model. The conclusions, highlighted in the model shown in Figure 8, claim that Drs inhibits *JNK* signaling which is shown with some key experiments. Yet, this model also claims that *JNK* signaling drives tissue disintegration effects of RasV12 through induction of *MMP2*, yet *MMP2* is not upregulated until a very late time point and only to a modest degree. Moreover, they don't show that *MMP2* or Hid (another *JNK* target) are required for these events. Also, in some cases the data is not as robust as would be expected in the field; activation of *JNK* is shown with compelling reporter data and pBsk staining, but the gene expression for *JNK* targets is underwhelming, for example Puc. Perhaps the issue is more with the model figure over-extending beyond the actual data presented.

In addition, the authors claims have relied mostly on qualitative nature of the data, whereas quantification could have strengthened their claim. For example, authors conclude that endoreplication leads to increase in nuclear volume but have not shown any quantitative increase in nuclear DNA content. The DAPI data shows most of the nuclei as disintegrated which will amount to higher nuclear volume but is less informative about DNA content. Further, authors have not stated whether the disintegrated nuclei were excluded from the analysis or not. Similarly, RasV12 stimulated MMPs-dependent tissue disintegration and Hid-dependent nuclear disintegration are not directly demonstrated.

Apart from the above-stated experimental concerns, issue with data presentation also significantly dampens the enthusiasm for this study. The data-presentation is incredibly cumbersome; throughout, the text does not present clearly the rationale of their experimental design, some key figure panels are not even mentioned in the text, other key data is buried in the supplemental figures, while other data is replicated in separate figure panels without clear reasoning, and importantly most of the legends/labeling on the graphs are so small, they are illegible. This last issue makes this work nearly impossible to critically evaluate. Also, I would encourage the authors to use gene names and symbols on their figures that are understood by a wide readership. For example, Bsk might be alternatively *JNK* in S5.

Reviewer #3:

This article is interesting because it shows that some immune genes, mostly antimicrobial peptide (AMP) genes, are induced in a highly artificial system in which the overgrowth of salivary glands is induced by ectopic expression of the Ras[V12] gain-of-function allele. The expression of immune genes would rely solely on Dorsal and not the Toll pathway (see below). The most important point is that the overexpression of Drosomycin is sufficient to reduce the induction of *JNK*-pathway regulated genes that occurs in the distal part of salivary glands, including pro-apoptotic genes, thus introducing the concept of a direct or indirect intracellular signaling function for an antimicrobial peptide.

In the longer term, it will be interesting to determine whether Dorsal plays a role in wild-type larval salivary glands and why it is differentially repressed in the DP.

Substantive concerns

1) The authors use RNAi lines to determine whether the Toll pathway is required for the induction of immune genes, which are not even validated. Since it is close to impossible to demonstrate the generation of a null phenotype by RNAi, the authors should imperatively use at least one null mutant of the Toll pathway, e.g., MyD88 and possibly spz.

2) The authors should look at and quantify Dorsal nuclear localization in the salivary glands and not solely at its expression. Upon looking at Figure 2B, one gets the impression it is nuclear only in the very proximal region of the organ.

3) Multiple AMPs are induced in the PP and it is surprising that the one AMP they have overexpressed is having such a biological function. A control is missing, ideally a Drosomycin mutant in which disulfide bridges cannot form; at least a nonrelevant gene should be used as a control to exclude nonspecific effects linked to overexpression in the Ras[V12] context. Also, the Drosomycin RNAi line does not abolish the expression of its target gene and does not have such a "drastic" effect: several genes are only mildly more strongly expressed. It would be interesting to use a Drosomycin null mutant. Perhaps less important for this work but of interest, it would have been informative to test mutants affecting AMP families that have recently been published.

4) It is not clear whether Drosomycin is secreted in the lumen or basally in the hemolymph. To really conclude that this is a tissue-autonomous response, inasmuch the authors do not state whether Drosomycin is expressed in other tissues, it would be useful to overexpress Drosomycin in the fat body and test its effects on the DP of the salivary glands.

---

## [Author Response]

[Editors’ note: the authors resubmitted a revised version of the paper for consideration. What follows is the authors’ response to the first round of review.]

Reviewer #1:This work proposes that tissue-autonomous Drosomycin (Drs) negatively regulates JNK signalling in a hypertrophic salivary gland (SG) model. The flow of experiment ideas is thorough, but lacking critical control treatments. There appears to be a spatial bias in the BxGal4 expression model used. Additionally, there is no demonstration that this is Drs-specific, as the authors never test for the effect of Gal4 dilution by involving additional UAS constructs such as other AMPs or a control like UAS-GFP. As the paper relies entirely on a BxGal4>UAS model, alternate Gal4 and UAS control treatments are essential to avoid spatial bias and the effect of multiple UAS diluting RasV12 production and consequent hypertrophy.List of substantive concerns:1) The authors use a BxGal4 model to drive SG expression throughout this manuscript. The assumption seems to be that BxGal4 is a pan-SG driver, and that differences in the proximal and distal parts of the SG arise due to differences in Drs expression in these regions. However it is revealed in Figure 2B that BxGal4 does not efficiently drive dorsal-RNAi in the proximal part (PP) of the SG, as this tissue retains abundant dorsal expression.

In order to address this very valid aspect systematically, we devised a novel algorithm to aggregate experimentally acquired fluorescence intensities across the vertical axes for every position along the longitudinal axis of salivary glands (Figure 1—figure supplement 1A-B; Author response image 1). This allowed to quantify Gal4-driver expression (i.e., BxGal4>UAS-mCD8::RFP), fluorescence signals from stainings (i.e., Phalloidin) or reporter lines (i.e., DrsGFP) along the salivary gland beyond a bipartite proximal-distal separation. To avoid artifacts due to differential thickness, tissue curvature and thus internal reflection we normalized BxGal4>UAS-mCD8::RFP signal intensities with Phalloidin signal from the corresponding positions and averaged for every genotype-time combination (e.g., *Ras^V12^* at 96 h) over more than 10 samples (i.e., 11-15).

**Author response image 1. sa2fig1:** *Ras^V12^*-glands show intrinsic anterior-posterior separation. (**A**) Right: Schematic representation of deployed algorithm to measure fluorescene signals along all vertical axis perpendicular to the glands longitudinal axis. Left: *mCD8::RFP;;Ras^V12^*- and *mCD8::RFP* -control glands stained with Phalloidin (green) to assess Bx^MS1096^ driven, Phalloidin-normalized RFP-expression at 96 h and 120 h AED along the longitudinal gland axis (middle). Signals averaged across n=15 glands showed 86% of the signal intensity in the proximal gland part (PP) compared to the distal (DP) (right). DrsGFP-reporter intensities along the longitudinal gland axis in *Ras^V12^*- and *w^1118^*-control glands at 96 h and 120 h AED (bottom). (**B**) Upper: Schematic representation of the correlation assay. DAPI-signal defines nuclei location and volume used to subsequently accumulate fluorescence signals from nuclear DrsGFP-reporter and dl-staining. Lower: Scatterplot of associated dl- and Drs-signals per nucleus in PP or DP across 5 glands at 96 h AED. Pearson correlation coefficient shown for all 5 samples. Lines with confidence intervals indicate linear regression for Drs- as a function of dl-signal per sample and gland compartment. (**C**) AttD gene expression as determined by qPCR in separated PP and DP of *Ras^V12^*- and *w^1118^*-control glands at 96 h and 120 h AED (log2-transformed, fold-change over Rpl32). Lower/upper hinges of boxplots indicate 1st/3rd quartiles, whisker lengths equal 1.5*IQR, red circle and bar represent mean and median. Student's t-tests determine statistical significance (** p<0.01, n.s. p≥0.05). (**D**) Ras^V12^-, *lgl^RNAi^*;*Ras^V12^*- and *w^1118^*-control glands carrying the 10xStat92E-GFP reporter (green) at 96 h and 144 h AED (*w^1118^*-larvae pupate at 120 h AED). (**E**) DrsGFP-reporter signal in *Ras^V12^*-glands at 144 h AED. (F) Expression of AMP genes in *Ras^V12^*, *lgl^RNAi^*;*Ras^V12^* or *lgl^RNAi^;Ras^V12^-PP*- in comparison to *w^1118^*-samples. Missing 1009 bars indicate absence of expression values in the RNAseq data. Insets: (**D-E**) DAPI. Scalebars: (**D-E**) 100 µm.

While Phalloidin-normalized BxGal4-expression increases between 96 and 120 h, no significant bias can be detected for either gland compartment (i.e., proximal or distal) in control glands (BxGal4,UAS-mCD8::RFP x *w1118*) apart from a minor tendency for increased BxGal4-driven expression in the proximal part at 120 h AED (Figure 1—figure supplement 1A-B; Author response image 1). Upon co-expression of *Ras^V12^*, the RFP-signal at 120 h compared to control glands increases further across the entire gland length. The proximal tendency from control glands is only reversed in *Ras^V12^*-glands at 120 h with a drop in BxGal4-expression along the proximal part (Figure 1—figure supplement 1A-B; Author response image A). This shift coincides with the differential replication of genomic DNA (i.e., endoreplications) in distal vs. proximal nuclei. However, while the differing number of endoreplications would suggest a substantially differing expression, on average the BxGal4driven expression in the proximal part of *Ras^V12^*-glands still reaches 86% of the average expression in the distal part (Figure 1—figure supplement 1A-B; Author response image 1). In addition, the expression strength in the proximal part still increases upon co-expression of *Ras^V12^* compared to control glands (*Bx, RFP* x *w1118*: 1.58; *Bx, RFP* x *Ras^V12^*: 2.30).

In parallel, quantifying DrsGFP-fluorescence across the length of the gland showed that Drs-expression and BxGal4-expression are neither correlated, nor anti-correlated over time (i.e., 96 and 120 h) and space (i.e., proximal and distal; Figure 1—figure supplement 1A-B; Author response image 1). Thus, while *Ras^V12^*-expression is qualitatively necessary for Drs-expression, already the comparably low BxGal4-driven *Ras^V12^*-expression at 96 h across the gland epithelium is sufficient for Drs to become fully expressed (i.e., same level as at 120 h AED). This makes Drs-expression qualitatively, but not quantitatively dependent on BxGal4-driven *Ras^V12^*-expression. In this regard, strong Drs-expression also precedes the accumulation of programmed cell death (PCD) in the gland as indicated by CC3 and nuclear disintegration (Figure 6C-F), a prerequisite for the model we propose, in which Drs actively inhibits induction of PCD.

In contrast, the intrinsic expression pattern of Dorsal, that appears independent of *Ras^V12^*-expression correlates not only qualitatively, but crucially quantitatively with the presence of Drs (Figure 2B-C). Under physiological conditions and thus independent of *Ras^V12^*, Dorsal expression is higher in the proximal part at 96 and 120 h AED than in the distal and eventually lost almost entirely in the distal part at 120 h (Figure 2B-C). This expression pattern persists upon *Ras^V12^*-coexpression. However, upon dl-knock-down without *Ras^V12^*-expression, Dorsal is indeed greatly reduced in all gland parts across the two developmental time points. We included separating lines in Figure 2B. to emphasize this aspect further. At 96 h, when Bx driven expression is similar in proximal and distal, residual dl-expression in the proximal part upon *dl^RNAi^*-expression indicates an insufficiency of the knock-down construct to abolish dl-expression (Figure 2B-C). This is in line with qPCR-data we acquired for Drs-expression in separated proximal and distal gland parts upon co-expression of the *dl^RNAi^*-line with *Ras^V12^* (Figure 2—figure supplement 3A; Author response image 3). However, in spite of a reduced Bx-driven expression in the proximal part of *Ras^V12^*-glands at 120 h and an insufficient *dl^RNAi^*-construct, dl-knock-down not only decreases Dorsal expression, but crucially also Drs-expression in both gland parts, as well as both time points (Figure 2B) emphasizing the dependency of Drs on dl rather than changes in Bxdriver expression further.

**Author response image 2. sa2fig2:** Drs asserts specific function on JNK-signaling not mimicked by other detected AMPs. (**A**) SG size as measured by outlining images of the 120 h-old, experimental and control glands with the indicated genotypes. (**B**) Quantified CC3-staining in *Ras^V12^*-glands with coexpressed Drs or mCD8::RFP as a UAS-dilution control, normalized for gland size. (**C**). Quantifications of activated *jnk* assessed by antiphosphoJNK-antibody staining and normalized for gland size in *Ras^V12^*-glands coexpressing Drs or mCD8::RFP. (**D**) Prototypical examples of Dronc- and *JNK*-activation in *Ras^V12^*-glands with coexpressed Drs or mCD8::RFP. (**E**) Expression values for Drs and *JNK* target genes as determined by qPCR for *Ras^V12^*-glands coexpressing either Drs or mCD8::RFP as a UAS-dilution control at 96 h and 120 h AED (*log2*-transformed; normalized to *Rpl32* expression). (**F**) *CecA1*-mRNA detected by *in-situ* hybridization in *Ras^V12^*- and *w1118*-control glands at 96 h and 120 h AED upon coexpression of either a *dl^RNAi^*-knock-down or a *CecA1*-overexpression construct. (**G**) Effect size of Drs-knock-down on Drs-expression in *Ras^V12^*-glands. Mean *Ras^V12^*-expression was set to “1” separately at 96 h and 120 h AED. (**H**) ISH with Drs-probe in *Ras^V12^*-glands with or without coexpressed *Drs^RNAi^*-construct. Scalebars: (D/F/H) 100 µm. Insets: (**D**) DAPI. Boxplots: (**A-C,E,G**) lower/upper hinges indicate 1^st^/3^rd^ quartiles, whisker lengths equal 1.5*IQR, red circle and bar represent mean and median. Significance evaluated by Student's t-tests (*** *p<0.001*, ** *p<0.01*, * *p<0.05, n.s. p≥0.05)*.

**Author response image 3. sa2fig3:** Apart from dorsal Drs is putatively regulated by Mef2. (**A**) Effect sizes of *dl^RNAi^*-knock-down on Drs-expression in PP- and DP-compartments of and compared to *Ras^V12^*-glands at 96 and 120 h AED. Mean of Drs-expression in *Ras^V12^*-glands was set to “1”. Boxplots: lower/upper hinges indicate 1st/3rd quartiles, whisker lengths equal 1.5*IQR, red circle and bar represent mean and median. (**B**) Transcription factor binding site analysis in up- and down-regulated genesets of *Ras^V12^*-, *lgl^RNAi^;Ras^V12^*- and *lgl^RNA^i;Ras^V12^-*PP- samples. Normalized enrichment scores represent statistical overrepresentation (NES>2) of transcription factor binding motifs in the 6 screened genesets. Cis-BP-derived motifs were sorted into the displayed transcription factor groups. NFκB- and jra- as well as Myb-, eg- and br-motif groups were included as positive and negative examples. (**C**) DrsGFP-reporter based assay including screened RNAi-lines targeting transcription factors identified in binding site analysis. 3 different phenotypes were scored (right) per genotype and their mean and standard deviation plotted. Dunn’s post-hoc test conducted on “None”phenotype to validate significant differences *(*** p<0.001, n.s. p≥0.05*). To avoid false positives, melanisation was monitored in pupae (insets) to confirm *Ras^V12^*-expression (Hauling et al., 2014). (**D**) Wider Drs-locus on “3L”- and “Dif/dl”-locus on “2L”-chromosome shown including outlines for gene loci encoded on plus and minus strands. Log-transformed motif scores are represented as a function of the motif’s location along the respective gene loci. (**E**) Drs in-situ hybridization in 120 h-old *Ras^V12^*- and control-glands hetero- or homozygous mutant for Myd88 (Myd88KG03447). Scalebar corresponds to 100 µm.

To confirm a correlation between the intrinsic, tissue-autonomous and largely *Ras^V12^*-independent dl-expression on the one and Drs-expression patterns in these hypertrophic glands on the other side, we quantified nuclear dl- parallel to DrsGFP-signal in 120 h-old *Ras^V12^*-glands. In fact, Drs and dl show a strong linear correlation across the entire scale (i.e., log_10_-transformed) in both the proximal as well as the distal gland part (Figure 2—figure supplement 2B; Author response image 1).

In Figure 4A there is a weakening of Drs signal in situ in the BxGal4>Drs at the PP, indicating BxGal4 is not efficiently driving Drs in the PP.

While ISH can be performed in a controlled, quantitative manner by using fluorescence probes and confocal microscopy, the ISH results presented throughout the manuscript were done by chromogenic staining and results were captured by brightfield microscopy. Thus, all results can only be seen in a comparative, semi-quantitative manner with the aim to characterize qualitative expression patterns. In addition, increased curvature and lumen width, as well as reduced tissue thickness and cell size of the proximal gland part complicate using ISH-quantifications as proxy for expression strengths further.

While we therefore refer to the presented quantifications along the longitudinal axis (Figure 1—figure supplement 1A-B; Author response image 1), we certainly acknowledge the point raised by the reviewer of a reduced expression in the proximal part of *Ras^V12^*-glands at 120 h, but present evidence that this reduced expression is not causal for the differences seen between proximal and distal gland compartments such as Drs-expression. To credit the reviewer’s comments and for full transparency, Figure 1—figure supplement 1A was included in the manuscript and its consequences for the model discussed in the manuscript’s main text.

Finally, the PCA analysis of RNAseq data shows totally independent clustering of the PP transcriptome from whole SGs; no BxGal4>w1118 PP control is given for comparison.

PC1 explains 94.26% of the variability in the data and all samples derived from overexpressing *Ras^V12^* (i.e., *Ras^V12^*, *lgl^RNAi^;Ras^V12^*, *lgl^RNAi^;Ras^V12^*-PP) are falling almost completely in line with one another in PC1 (Figure 3A). Thus, while all 9 *Ras^V12^*-samples separate in PC1 from the control gland samples with the majority of variability in the expression data being explained by PC1, the *lgl^RNAi^;Ras^V12^*-PP samples separate mainly in PC2, which only explains 3.05% of the intrinsic variability of the data.

In addition, principal components analysis subsumes qualitative differences (i.e., genesets expressed) into quantitative differences (i.e., gene expression intensities), indicating that the separation in PC2 remains largely due to qualitative differences in significantly differentially expressed genes that do not occur in the DP. This is clearly underlined in the barplot (UpSetR-plot) in Figure 2A, which shows a set of 309 genes solely, significantly differentially expressed in the PP and further exemplified in Figure 2C. for “immune response” genes. Many of these genes show a strong qualitative difference (i.e., unilateral upregulation in the PP compared to whole-tissue samples). The whole-tissue samples include not only the distal, but also the proximal part, which dampens the detectable difference in gene expression strength between both parts. Thus, we performed additional qPCRs for AttD as the highest upregulated gene in the PP on dissected, separated gland parts (DP vs. PP), which revealed a complete absence of AttD-expression in the distal part at 96 and 120 h (Figure 3—figure supplement 1D; Author response image 1).

In summary, the qualitative differences between the PP and the DP as reflected in the expression of different sets of genes (Figure 2A-C) gives rise to the separation of the PPsamples along PC2 which in turn however only explains a minor fraction of the data’s inherent variability.

This spatial separation is further corroborated by the expression patterns shown for amongst others dl (Figure 2B), DrsGFP (Figure 1E; Figure 1—figure supplement 1E; Figure 5A) and TREGFP1b (Figure 3F; Figure 3—figure supplement 1F; Figure 5F; Figure 5—figure supplement 2E). However, a reporter indicating JAK-STAT-activation (related GOterms were found to be enriched in the RNAseq-datasets; *data not shown*), the 10xSTAT92EGFP-reporter, highlights this separation best (Figure 3—figure supplement 1F; Author response image 1). Upon *Ras^V12^*-expression, the construct reports JAK-STAT-activation solely in the PP between 96 h to 144 h (Figure 3—figure supplement 1F; Author response image 1). The presented evidence for an intrinsic spatial separation into a proximal and a distal gland compartment is in the following also central for the proposed model, in which a dl-dependent reduction solely in the DP reverts inhibition of *JNK*-signaling.

Thus, throughout the manuscript, the PP is very likely a region with significantly reduced RasV12 expression, readily explaining most differences between the sick and apoptotic DP and the (relatively) healthy PP. For instance, this would readily explain differences in hemocyte attachment between the DP and PP. This may also imply a Drs response is seen in the PP solely because this relatively healthy tissue remains immune competent for longer than the DP.

If PCD and tissue degradation is the cause for the reduced Drs expression or expression in general, the assumption would be a maintained Drs-expression upon inhibition of PCD in the distal part of *Ras^V12^*-glands at 120 h. Upon inhibition of the *JNK*-pathway by co-expression of a *jnk^DN^*-construct (*bsk^DN^*), *JNK*-activation (Figure 5—figure supplement 1B) as well as hemocyte attachment (Figure 4D) are reversed almost to the level of control glands in line with the block of PCD. However, in spite of inhibited PCD, Drs-expression does not recover in the DP, ultimately rendering Drs-expression independent of the quantitative expression levels from the BxGal4-driver (Figure 4A; Figure 5—figure supplement 1A). Drs is only dependent on the driver in as much as it qualitatively expresses *Ras^V12^* in this model. In parallel, *jnk^DN^;;Ras^V12^*-glands undergo the same increase in endoreplications, thus mimicking the changes in Bx-driven expression present in *Ras^V12^*-glands (Figure 6C).

Moreover, Figure 5—figure supplement 2C. shows an upregulation of many *JNK* target genes upon co-expression of an RNAi-knock-down construct against Drs at 96 h, when the DP of *Ras^V12^*glands still expresses Drs and is thus susceptible to a Drs-knock-down. Despite the use of two UAS-constructs, an increased expression of these *JNK* target genes indicates that a Drs-reduction has a specific, active effect on the expression of the *JNK* target genes.

In summary, the state of a gland part (i.e., healthy or not) is not determined by the strength of Bx-driven expression, but instead by the presence or absence of Drs, whose expression is in turn regulated by the distribution of dl.

Finally, Drs expression in the SG increases between the L3 and prepupa stages of development (FlyAtlas). It is unclear how BxGal4>RasV12 might affect SG developmental patterning, and indeed the model relies on rapid overgrowth of the tissue. Moreover there is an expected Drs response in immune-responsive tissues to sterile injury or stress. It is likely that SG remodelling during RasV12-induced stress leads to corresponding programmed cell death and similarly induction of NF-κB. This interpretation is congruous with the findings of Araki et al. (2018) and Parvy et al. (2019) that suggest NF-κB effectors cooperate in clearing tumorous tissue and responds to cell death signals.

The model presented in this manuscript and the reviewers alternative fundamentally differ in their interpretation of the causality between Drs-activation and the induction of PCD. (1) Either Drs reduced cell death by inhibiting *JNK*-signaling or (2) PCD leads to the erasure of Drs-expressing cells and thus eventually to Drs-expression itself. While we acknowledge the 14%reduction in the PP compared to the DP of *BxGal4*>*Ras^V12^*-glands at 120 h, the alternative proposition can however not be sufficiently aligned to all the above presented evidence at once: (1) no correlation between strength of Bx-driven and Drs-expression (Figure 1—figure supplement 1A-B; Author response image 1), (2) Drs-expression precedes occurrence of PCD (Figure 1—figure supplement 1A-B; Author response image1A), (3) Drs-expression is determined by dl (i.e., dl-reduction leads to Drs-reduction; Figure 2B; Figure 2—figure supplement 2B; Figure 2—figure supplement 3A; Author response image 1; Author response image 3), (4) tissue-autonomous proximal-distal-separation of the gland epithelium (Figure 1E; Figure 1—figure supplement 1G,I; Figure 2B; Figure 3F; Figure 3—figure supplement 1F; Figure 5F; Author response image 1) and (5) no recovery of Drs-expression by blocking PCD via *JNK*-inhibition (Figure 4A; Figure 5—figure supplement 1A).

Most importantly, Drs is not a mere bystander parallel to the tissue’s disintegration, but has functional relevance in preventing it as is shown by Drs-overexpression in *Ras^V12^*-glands. Not only are markers for PCD reduced (Figure 6D. – nuclear disintegration; Figure 6E-F. – CC3-activation; Figure 5E-F. – *JNK*-activation) and thus tissue-viability maintained, but the glands continue to overgrow (Figure 4C; Figure 4—figure supplement 1A; Author response image 2). In this regard, the here presented model of a tissue-autonomous immune response aiding hypertrophic growth is also opposed to the models published by Araki et al. and Parvy et al. (Araki M et al. Dis Model Mech. 2019 Jun18;12(6):dmm037721. doi:10.1242/dmm.037721; Parvy JP et al. *eLife*. 2019 Jul 30;8:e45061. doi:10.7554/*eLife*.45061). Clearance of the distal parts of *Ras^V12^*-glands at 120 h not only occurs in the absence of Drs, but in its presence such as by Drs-overexpression, no gland tissue needs to be cleared, since Drs prevents its disintegration in the first place.

Kenmoku et al. indeed showed the induction of Drosomycin upon sterile injury independent on the position of the actual injury in the larval epidermis (i.e., pinching; Kenmoku H et al. Dis Model Mech. 2017 Mar1;10(3):271-281. doi:10.1242/dmm.027102). However, this type of sterile injury leads exclusively to an upregulation of Drs in the fat body as part of a systemic immune response. While we use the same transgenic reporter construct as Kenmoku H et al. we exclusively identify expression of Drs in the larval salivary glands coinciding with *Ras^V12^*-overexpression, indicating the absence of artefacts due to mechanically induced stress (Ferrandon D et al. EMBO J. 1998 Aug10;17(5):1217-27. doi:10.1093/emboj/17.5.1217; Jung AC et al. Biotechniques. 2001 Mar;30(3):594-8, 600-1. doi: 10.2144/01303rr04). In parallel, neither are larval salivary glands established immune responsive epithelia, nor did other *Drosophila* overgrowth models show expression of immune effectors in a tissue-autonomous manner. Apart from the qualitative differences in tissue expression, we also identify strong quantitative differences in the Drosomycin expression levels with respect to the reported levels upon sterile injury. While Kenmoku report a fold change of maximal 10 in Drs expression upon injury over control levels (i.e., in fat bodies), Drs expression in *Ras^V12^*-glands over control glands is two orders of magnitude higher (i.e., log2(FC) > = 10, FC > = 1024).

Apart from our findings presented here, Pierce SB et al., 2004 showed a similar proximal, distal separation of dMyc and dMnt expression along the gland epithelium in larval salivary glands (Pierce SB et al. Development. 2004 May;131(10):2317-27. doi:10.1242/dev.01108). In addition, our findings on the differential expression and abundance of dl in the absence of *Ras^V12^* (Figure 2B) and the differing nuclei volume in PP and DP consistent with previous reports indicate that the SG pattern identified in *Ras^V12^* glands is not altered, but that the *Ras^V12^*-related phenotypes are indeed based on an intrinsic separation between PP and DP that is pre-established in wandering stage larvae (Berendes HD, Ashburner M 1978, The Salivary Glands. The Genetics and Biology of *Drosophila*, vol. 2b, pp. 453-498. London: Academic Press; Hammond MP, Laird CD. Chromosoma. 1985;91(3-4):279-86. doi:10.1007/ BF00328223).

2) Throughout the manuscript, the authors combine additional UAS constructs without controlling for the dilution of their Gal4 drive. This affects almost every figure in the latter half of the manuscript, and the effect of this dilution is apparent in Figure S5C, where Drs levels in BxGal4>Drs alone are 1-2 fold higher than BxGal4>Drs,RasV12 double drive.

Previously, we co-expressed a cytoplasmic GFP alongside *Ras^V12^*, which did not diminish any of the reported effects and phenotypes induced by *Ras^V12^* alone (Figure 2 and Figure 3 in Hauling T et al. Biol Open. 2014 Apr15;3(4):250-260. doi:10.1242/bio.20146494). However, to address this aspect more systematically, we used a mCD8-tagged RFP-line to balance the number of UAS constructs with UAS-Ras^V12^ and UAS-Drs. Crucial experiments including the activation of *JNK* (Figure 5—figure supplement 3B-C; Author response image 2) and CC3 (Figure 6—figure supplement 1C-D; Author response image 2,D), as well as the expression of *JNK*-target genes (Figure 5—figure supplement 3A; Author response image 2) were repeated, while accounting for the number of UAS-constructs. Consistent with our previous results, *Drs,Ras^V12^*-glands exhibit significantly reduced CC3- and pJNK-signal compared to *Ras^V12^*-samples, while *RFP;;Ras^V12^*-glands do not differ significantly from *Ras^V12^*-glands (Figure 5—figure supplement 3B-C; Figure 6—figure supplement 1C-D; Author response image 2). Most importantly, while *Drs,Ras^V12^*-glands show a drastic increase in gland size emphasizing the hypertrophic overgrowth, co-expression of *RFP* alongside *Ras^V12^* has no significant influence on the size of the glands (Figure 4—figure supplement 1A; Author response image 2). This indicates a physiological influence of Drs that is not replicated by simply co-expressing a second protein such as RFP.

Quantifications of *JNK*-target gene expressions corroborated the difference between Drs- and RFP-co-expression further. At 96 h, apart from *Mmp2* all monitored target genes show no significant differences in expression between *RFP;;Ras^V12^*- and *Ras^V12^*-glands, while for the same genes *Drs,Ras^V12^*-glands exhibit significantly reduced expression compared with *Ras^V12^*-glands (Figure 5—figure supplement 3A; Author response image 2). At 120 h, Mmp1, *Mmp2* and upd1 show the same pattern with these genes being significantly downregulated in *Drs,Ras^V12^*-, but not in *RFP;;Ras^V12^*-glands compared to *Ras^V12^*. While the absolute reduction in gene expression is still stronger in *Drs,Ras^V12^*-glands for *puc*, *hid* and *rpr*, their *RFP;;Ras^V12^*-counterparts show a significant decrease, too (Figure 5—figure supplement 3A; Author response image 2).

While we acknowledge the partial reduction of these genes in *RFP;;Ras^V12^*-compared to *Ras^V12^*-glands, several confounding factors complicate the comparability of results from co-expressed constructs used as controls: (1) insertion site of the UAS-constructs, that have an influence on their expressability, (2) metabolic burden due to the size and cellular location of the final protein, in line with the reviewers comment (see below), (3) activity and function of the protein and (3) whether the protein is secreted or not.

Since Drs is a secreted peptide, we opted for mCD8::RFP as control to mimic the entrance into the secretory pathway, too. However, while mCD8::RFP-constructs were shown to be inert in other cell systems, their influence on salivary gland physiology with e.g., its particular high secretory activity has not been documented.

Moreover, sole expression of *Ras^V12^* and co-expression of *Ras^V12^* with Drs show very similar Drs-expression levels, both at 96 h and 120 h AED. In addition, as indicated by the reviewer, the co-expression of *Drs^RNAi^* with *Ras^V12^* results in a strong reduction of Drs expression (i.e., only 8% at 96 h and 4% at 120 h of Drs-expression remains after *Drs^RNAi^*-co-expression; Figure 5—figure supplement 2A; Author response image 2), and thus shows the opposite effect as in *Drs,Ras^V12^*-glands (Figure 5—figure supplement 2CD; Figure 6A). In this regard, the most parsimonious explanation for the differences in expression are the actual constructs driven by BxGal4 together with *Ras^V12^* (i.e., alone or in combination) and their differing downstream effects. In fact, the RNAseq data for *Ras^V12^* (Figure 1D; Figure 3A-C) is consistent with this explanation highlighting the comprehensive reprogramming, the glands undergo. Moreover, the co-expression of another RNAi-construct targeting *lgl* increases the severity of the phenotypes (i.e., nuclear and tissue disintegration) further (Figure 1—figure supplement 1C-H).

We also followed the reviewer’s suggestion to overexpress one of the other AMPs upregulated in *Ras^V12^*-glands, CecA1 (4^th^ highest upregulated; Figure 3C). ISH with a CecA1specific probe revealed an expression of CecA1 exclusively in the PP of *Ras^V12^*-glands at 96 and 120 h AED in line with the RNAseq-results (Figure 3C; Figure 3—figure supplement 1E; Author response image 2). By using the same probe, overexpression throughout the whole gland epithelium at both time points with the help of the CecA1-overexpression construct could also be confirmed. However, *CecA1;Ras^V12^*glands do not exhibit the same abundant CecA1-expression as in *CecA1*-glands, but instead CecA1-expression is confined to the PP. While the implications of a *Ras^V12^*-dependent inhibition of CecA1-transcript accumulation in the distal part are beyond the scope of this manuscript, it is obvious that CecA1-coexpression in methodological terms is not simply controlling for Drs-co-expression. In fact, this result suggests further layers of direct interactions between *Ras^V12^*-driven hypertrophy and the tissue-autonomous immune response beyond immediate expectations.

If RasV12 is diluted in combined UAS lines, this would promote a delayed phenotype, resulting in an apparent rescue effect if you simply compare equal time points. But it would be expected that continued monitoring to 144h would reveal the same apoptotic effects as the lone BxGal4>RasV12 tissue at 120h, similar to how BxGal4>RasV12 at 96h displays less apoptosis than 120h.

We have included a prototypical example of a *Ras^V12^*-gland at 144 h AED. At this time point, Drs is not only still expressed, but remains confined to the PP reminiscent of its expression at 120 h (Figure 1—figure supplement 1I; Author response image 1).

In addition, a model of gradual induction of PCD and its delay in *Drs,Ras^V12^*-glands or other genotypes due to dilution effects implies the reduction of expression of all genes in the gland including Drs due to apoptotic induction. However, given the necessity for JNK signaling during the induction of PCD in *Ras^V12^*-expressing glands, the inhibition of JNK signaling throughout the entire time of *Ras^V12^*-expression rescues the glands from undergoing PCD (Figure 6A,D,E-F). However, the glands continue to show a difference in Drs expression between proximal and distal part with a continuous loss of Drs expression in the latter at 120 h AED (Figure 4A; Figure 5—figure supplement 1A).

There is one argument that can be made in favour of the dilution not mattering: Drs-RNAi has a mild but opposite effect at 96h. However it should be noted that RNAi vs. protein overexpression leads to significantly different metabolic burdens on the cell, and thus RNAi is not a good control for this dilution effect. The authors should use a control overexpression construct such as UAS-GFP or UAS-RFP throughout.

We hope that the above outlined additional experiments based on the co-expression of especially mCD8::RFP as well as CecA1 combined with the accompanied clarifications emphasize the unique role Drs has in this hypertrophic model and that its significance is not merely explained by dilution effects. In this regard, we also like to emphasize the significant differences upon *Drs^RNAi^*- and *Drs*-co-expression with *Ras^V12^* and the *log2*-scale used to represent associated qPCR expression values, which can influence the impression of expression differences.

3) There is no logic presented within the paper that this effect should be Drs-specific. Many other AMPs and immune effectors were similarly upregulated in their transcriptome (e.g. AttD, Def). If AttD and Def are similarly induced in this model, it is assumed their expression is similarly regulated, and that overexpression of AttD or Def could similarly rescue RasV12 SGs, and RNAi of AttD or Def could similarly exacerbate RasV12 SGs. I would recommend that, alongside a UAS-GFP control, the authors also test at least some subset of NF-κB effectors alongside their Drs model to elucidate if this is specific to Drs or a general effect. Given findings in other papers on AMP-tumor interactions where overexpressing Drs, Dpt, or Def all had similar effect (Araki et al., 2019), it seems likely that many AMPs should result in the same phenotype, implying there is no molecular pathway where specifically Drs inhibits JNK.

Based on the acquired RNAseq-datasets, Drosomycin is the most highly upregulated gene in whole *Ras^V12^*-glands (Figure 1D). More importantly, Drosomycin is also the only AMP that is consistently highly upregulated in all three genotypes whose transcriptomes were sequenced (Figure 3—figure supplement 1C; Author response image 1). Whole tissue samples of *Ras^V12^*- and *lgl^RNAi^;Ras^V12^*-glands showed high expression for only one more AMP, AttD (Figure 3—figure supplement 1C; Author response image 1). However, precise transcript quantifications via qPCR on separated PPs and DPs revealed that this expression results from the upregulation of AttD solely in the PP (Figure 3—figure supplement 1D; Author response image 1). Crucially, this restricted upregulation not only occurs at 120 h AED as for Drs, but also at 96 h AED. Thus, Drs is the only AMP that is expressed at 96 h AED in the distal part of *Ras^V12^*-glands and whose downregulation coincides with an increase in *JNK*-activation and initiation of PCD from 96 h to 120 h AED. In this regard, Drs emerged as the only likely candidate to influence the shift in the DP, an aspect we definitely have missed to clarify in the manuscript before, but have added now.

However, our intention was to merely indicate that Drs expression is capable to interfere with *JNK*-signaling and sufficient to inhibit it in the presented hypertrophy model, but we did not comment on a unique function of Drs in *JNK*-inhibition. In this way, we certainly acknowledge the possibility of additional AMPs having similar or different effects on stress signaling in this or other models of overgrowth. However, the mechanism of Drs directly interfering with *JNK*-signaling rather than with the accumulation of initial stress is a novel concept shedding new light on the necessity of tissue-autonomous immune responses on tissue integrity.

As a proof-of-principle, we overexpressed CecA1 alone or in combination with *Ras^V12^* leading to a block of CecA1-transcript accumulation in the DP of *Ras^V12^*-glands (Figure 3—figure supplement 1E; Author response image 2). While the underlying mechanism resulting in this block awaits further investigations, it highlights the role of Drs as the only AMP expressed in the DP until 96 h AED further.

Reviewer #2:Krautz et al. presents an interesting finding on regulation of stress signaling during hypertrophy via tissue-autonomous immune signaling. Data shows that Drosomycin, downstream of the NFkB factor Dorsal, interferes with overgrowth of the salivary gland caused by RasV12 expression. The authors investigate the underlying mechanism of this finding using different approaches, presenting copious data which are generally supportive their model.However, in some cases the findings lack direct demonstration of key aspects of their model. The conclusions, highlighted in the model shown in Figure 8, claim that Drs inhibits JNK signaling which is shown with some key experiments. Yet, this model also claims that JNK signaling drives tissue disintegration effects of RasV12 through induction of MMP2, yet MMP2 is not upregulated until a very late time point and only to a modest degree.

According to the acquired qPCR-data *Mmp2* is for the first time significantly differentially expressed at 120 h AED (Figure 4D; p=0.0263; FC=2.60; log_2_(FC)=1.38). However, while the upregulation is not significantly changed compared to *w1118*-control glands, *MMP2* is on average already differentially expressed in *Ras^V12^*-glands at 96 h (Figure 4D; p=0.2928; FC=2.15; log2(FC)=1.10). Given the time necessary for *JNK*-dependent transcriptional activation around 96 h AED, generating and secreting the *Mmp2*-protein, as well as enzymatically disrupting the basement membrane and consequentially recruiting hemocytes, the presented data is consistent with the model proposed in the manuscript. While a non-significant, but differential Mmp2expression already at 96 h AED indicates an incline in expression over time, the detection of *Mmp2*-dependent disruption of the basement membrane at 120 h characterizes the functional consequences in a time-appropriate manner (Figure 4D).

Moreover, overexpression of Mmp1 or *Mmp2* independently of one another and of *Ras^V12^* indicates that *Mmp2* is the only matrix metalloproteinase capable of opening the BM surrounding the glands irrespective of strong differences in the expression of Mmp1 and *Mmp2* in *Ras^V12^*-glands (Figure 4E; Figure 4—figure supplement 1B).

Moreover, they don't show that MMP2 or Hid (another JNK target) are required for these events.

While the results in the manuscript covered presence (i.e., expression, Figure 4D) and functional capability (i.e., sufficiency, Figure 4E; Figure 4—figure supplement 1B) of *Mmp2* to degrade the gland’s basement membrane, we indeed failed to show the necessity of *Mmp2* for the recruitment of hemocytes as a function of basal membrane degradation in *Ras^V12^*-glands in line with the reviewer’s comment.

For this purpose, we crossed a *Mmp2*-mutant allele (*Mmp2^k00604^*) with *Ras^V12^* and BxGal4 to obtain a hetero- or homozygous knock-out of the *Mmp2*-locus (Figure 4F-G; Figure 4—figure supplement 1C; Author response image 4,D). Both, in a homozygous, but also already in a heterozygous mutant state, hemocyte attachment is significantly impaired compared to *Ras^V12^*-glands (Figure 4F-G; Figure 4—figure supplement 1C; Author response image 4,D). To capture the true extent of this reduction (i.e., hemocyte attachment measured as *ln*(hem-area)/*ln*(gland size)), we also plotted non-log-transformed effect sizes of hemocyte attachment in *Mmp2^-/+^;Ras^V12^*- and *Mmp2^-/-^;Ras^V12^*- compared to *Ras^V12^*-glands (Figure 4G; Author response image 4). In fact, in both mutant backgrounds, hemocyte attachment has dropped to 33% of the average quantified in *Ras^V12^*-glands. In addition, the remaining, attached hemocytes are far less spread, rounded and do not form filo- or lamellipodia as their counterparts on the surface of *Ras^V12^*-glands do (insets in Figure 4—figure supplement 1C; Author response image 4). Both, the significant reduction of attached hemocytes and the morphology of the residual, surface-bound hemocytes indicates the crucial involvement of *JNK*-dependent *Mmp2*-expression in recruitment and activation of hemocytes as a proxy for the systemic immune response towards *Ras^V12^*-glands.

**Author response image 4. sa2fig4:** Mmp2 expression necessary for hemocyte attachment to the surface of *Ras^V12^*-glands. (**A**) Hemocyte attachment at 120 h AED in *Ras^V12^*-glands hetero- and homozygous mutant for *Mmp2* (*Mmp2^k00604^*). (**B**) Effect size of hetero- and homozygous mutant *Mmp2*-allele (*Mmp2^k00604^*) on hemocyte attachment to and compared to *Ras^V12^*-glands. Mean of hemocyte attachment in *Ras^V12^*-glands at 120 h was set to “1”. E. and F. represent the same data points. (**C**) Quantification of CC3 to measure Dronc-activation upon *hid*knock-down in *Ras^V12^*-glands. (**D**) *w1118*-control and *Ras^V12^*-glands hetero- or homozygous for *Mmp2* (*Mmp2^k00604^*) stained for attached hemocytes with anti-Hemese-antibody. Lower image row shows magnifications of the insets in the upper row. Filopodia (filled arrowheads) and lamellipodia (open arrowheads) indicate hemocyte activation on the surface of *Ras^V12^*-, but not glands mutant for *Mmp2*. (**E**) H2DCF-staining of live glands as general means to evaluate the presence of ROS at 96 h and 120 h AED. (**F**) Dronc-activation upon *hid*-knock-down in *Ras^V12^*glands indicated by CC3-staining normalized by SG size per gland. Insets: (**D,F**) DAPI, (**E**) brightfield. Scalebars: (**D**. Upper,**E-F**) 100 µm, (**D**. Lower) 50 µm. Boxplots (**A-C**): lower/upper hinges indicate 1^st^/3^rd^ quartiles, whisker lengths equal 1.5*IQR, red circle and bar represent mean and median. Significance evaluated by Student's t-tests (*** *p<0.001*, ** *p<0.01*, * *p<0.05)*.

These findings are in line with previous reports on the role of *Mmp2* in cell dissociation, the integrity of basement membranes and hemocyte attraction (Jia Khan Q et al. Sci Rep. 2014;4:7535. doi:10.1038/srep07535; Kim MJ, Choe KM. PLoS Genet. 2014;10(10):e1004683. doi:10.1371/ journal.pgen.1004683). Moreover, recent, systematic work from the Bergmann lab has not only emphasized the role of *Mmp2* in hemocyte recruitment to undead cells and a clonally induced, neoplastic tumor model, but also showed the importance of reactive oxygen species (ROS) in this process (Diwanji N, Bergmann A. Nat Commun. 2020;11(1):3631. doi:10.1038/s41467-020-17399-8). In order to evaluate, whether ROS are also present in the here presented hypertrophic model and thus could aid in the process of basement membrane degradation, we stained live tissues with H2DCF-dye to detect a wide spectrum of ROS in salivary glands (Figure 4—figure supplement 1D; Author response image 4). In fact, while *Ras^V12^*-glands show a strong ROS-signal in the DP, co-expression of either *Drs* or *jnk^DN^* with *Ras^V12^* appears to prevent ROS-generation. This indicates that apart from inhibiting *Mmp2*-expression by blocking JNKsignaling, Drs might also block hemocyte recruitment to the gland surface by inhibiting ROS production through its block of *JNK*-activation and could explain the complete absence of attached hemocytes on *jnk^DN^;;Ras^V12^*- and *Drs,Ras^V12^*-glands in contrast to *Mmp2^-/-^;Ras^V12^*- and *Mmp2^-/+^;Ras^V12^*-glands (Figure 4D,F-G; Figure 4—figure supplement 1C; Author response image 4,D). However, an association between *JNK*-signaling and ROS-production in *Ras^V12^*-glands needs further investigations.

In parallel, to provide further evidence on the significance of hid in *JNK*-dependent salivary gland degradation, we coexpressed a *hid*-knock-down construct with *Ras^V12^* and stained for CC3 as an indicator for the induction of PCD (Figure 6—figure supplement 1E-F; Author response image 4,F). Upon reduction of *hid* expression, the downstream activation of Dronc is significantly reduced compared to *Ras^V12^*-glands. In fact, coexpression of *hid^RNAi^* reduces CC3 to the same level as upon coexpression of *Drs* with *Ras^V12^* indicating that hid plays a crucial quantitative role in the PCD blocked by Drs (Figure 6F; Figure 6—figure supplement 1E-F; Author response image 4,F).

Also, in some cases the data is not as robust as would be expected in the field; activation of JNK is shown with compelling reporter data and pBsk staining, but the gene expression for JNK targets is underwhelming, for example Puc.

Due to the high expression values identified for *JNK* target genes in *Ras^V12^*-expressing glands, we consistently plotted all expression data as *log2*-transformed values, which for instance implies puc is significantly increased in *Ras^V12^*-glands at 96 h over control glands. The presented qPCR values for puc are not only in line with the order of magnitude of previously reported puc expression, but comparable to *puckered* expression values after immune stimulation or initiation of regeneration (Hu Y et al. G3 (Bethesda). 2013 Sep 4;3(9):160716. doi: 10.1534/g3.113.007021; Chakrabarti S et al. PLoS Genet. 2016 May27;12(5):e1006089. doi:10.1371/journal.pgen.1006089. eCollection; Khan SJ et al. PLoS Genet. 2017 Jul28;13(7):e1006937. doi:10.1371/journal.pgen.1006937).

For instance, we obtained the sequencing data for *scrib-* and *dlg*-tumor models published in Bunker et al. and compared the fold-change for *puc*-expression with the values from the salivary gland models presented here (Author response image 5; Bunker BD et al. *eLife*. 2015 Feb26;4. doi:10.7554/ *eLife*.03189). Apart from an expected reduction in proximal gland parts (i.e., *lgl^RNAi^;Ras^V12^*-PP), our RNAseq data shows comparable *puc*-expression in *Ras^V12^*- and *lgl^RNAi^;Ras^V12^*-glands. Moreover, qPCR-data for *puc* (presented as non-log-transformed data for comparison) in *Ras^V12^*-glands at 96 h and 120 h, indicates that the actual *puc*-expression in hypertrophic glands is even higher (i.e., FC=3-4; Author response image 5). These values are above the induction level reported for *puc* during *JNK*-dependent wing disc pouch region regeneration and similar to *puc*-levels in gut, hemocytes and the fat body upon septic injury (Khan SJ et al. PLoS Genet. 2017 Jul28;13(7):e1006937. doi: 10.1371/journal.pgen.1006937).

**Author response image 5. sa2fig5:** Puc expression levels in *Ras^V12^*-glands comparable to other tumor models. (**A**) Differential expression values for puc as quantified from the here presented RNAseq datasets and Bunker, B.D. et al., 2015. Comparison of sequenced genotypes are presented in the respective bars. (**B**) Fold-change expression values for puc without log-transformation in RasV12-glands with Drs-overexpression (Drs) and -knock-down (DrsRNAi). Data reproduced from Figure 5—figure supplement 2C-D. (**C**) Effect sizes of puc and hid upon DrsRNAi-co-expression in RasV12-glands with respective mean expression in RasV12-glands set to “1” for each gene and time point. (**D**) Differential expression values for egr, grnd and wgn across the three sample populations screened with RNAseq (i.e., RasV12, lglRNAi;RasV12 or lglRNAi;RasV12-PP each compared to *w1118*-glands). Q-values are presented in the respective bars. (**E**) DrsGFP-signal normalized to SG size per gland in RasV12- and w1118control glands each separated by sex into female and male specimens. (**F**) Effect size for DrsGFP-signal in male RasV12-glands compared to their female equivalents with mean of female RasV12-glands set to “1”. Boxplots (**BC,E-F**): lower/upper hinges indicate 1st/3rd quartiles, whisker lengths equal 1.5*IQR, red circle and bar represent mean and median.

We have performed similar comparisons with reported and published *JNK*-target genes beyond *puc* (data not shown), which are similarly in line with the differential expression values identified in *Ras^V12^*-glands.

Perhaps the issue is more with the model figure over-extending beyond the actual data presented.

We hope that the additional evidence for *Mmp2* and *hid* being involved in the activation of a systemic immune response and degradation of the gland tissue, reformatting of the figures and a softened choice of words cover the aspects rightfully criticized by the reviewer.

In addition, the authors claims have relied mostly on qualitative nature of the data, whereas quantification could have strengthened their claim. For example, authors conclude that endoreplication leads to increase in nuclear volume but have not shown any quantitative increase in nuclear DNA content. The DAPI data shows most of the nuclei as disintegrated which will amount to higher nuclear volume but is less informative about DNA content. Further, authors have not stated whether the disintegrated nuclei were excluded from the analysis or not. Similarly, RasV12 stimulated MMPs-dependent tissue disintegration and Hid-dependent nuclear disintegration are not directly demonstrated.

We agree entirely with the reviewer’s notion on comprehensively quantifying phenotypes as a necessity for repeatable and transparent science. While several phenotypes throughout the paper were indeed demonstrated only in a qualitative, proof-of-principle manner (i.e., Mmp1/2overexpression; spatio-temporal dl-distribution), conclusions drawn from these instances were kept to their essential minimum (i.e., *Mmp2*, but not Mmp1 capable of disrupting the basement membrane; overlap of dl with Drs expression patterns). In parallel, phenotypes which the model relies upon were not only quantified, but validated by independent repetitions of the experiments (data not shown), including nuclear volume (Figure 1C; Figure 1—figure supplement 1D-E; Figure 6C), hemocyte attachment (Figure 1F; Figure 1—figure supplement 1H; Figure 4D), Drs-expression (semi-quantitatively as part of the regulator screen; Figure 2A; Figure2—figure supplement 2A), Drs- and *JNK*-target gene expression via RNAseq and qPCR (Figure 3C,E; Figure 3—figure supplement 2A; Figure 4E; Figure 5B; Figure 5—figure supplement 1B; Figure 5—figure supplement 2C-D; Figure 6A), SG-size (Figure 4C), phosphorylation/activation of *JNK* (Figure 5C; Figure 5—figure supplement 2F), *JNK*-reporter activation (Figure 5D; Figure 5—figure supplement 2E; Figure 7B,D) and Dronc-activation (Figure 6F; Figure 6—figure supplement 1A; Figure 7C; Figure 7—figure supplement 1B).

In the course of revising the manuscript, we included additional quantifications, i.e., BxGal4-driver strength (Figure 1—figure supplement 1A-B; Author response image 1), dl-Drs-correlation (Figure 2—figure supplement 2B; Author response image 1), *dl^RNAi^*- and *Drs^RNAi^*-effect sizes (Figure 2—figure supplement 3A; Author response image 3; Figure 5—figure supplement 2A; Author response image 2), Drs-expression (semi-quantitatively as part of the regulator screen, Figure 2—figure supplement 3C; Author response image 3), AttD-expression (Figure 3—figure supplement 1D; Author response image 1), SG-size (Figure 4—figure supplement 1A; Author response image 2), *JNK* activation (Figure 5—figure supplement 3B; Author response image2C), activated Dronc (Figure 6—figure supplement 1B-C; Author response image 2; Author response image4C), Drs- and *JNK*-target expression via RNAseq and qPCR (Figure 3—figure supplement 1C; Author response image 1; Figure 5—figure supplement 3A; Author response image 2) and hemocyte attachment (Figure 4F-G).

To characterize the impact of continuous Ras-MAPK-signaling, we assayed nuclear volume as an indicator of pre-/absence of additional rounds of endoreplication. In line with the reviewers comment, DAPI has been described as a less efficient fluorochrome for measuring DNA content in the past due to its heterogeneity in nuclear distribution (Santisteban MS et al. J Histochem Cytochem. 1992;40(11):1789-1797. doi:10.1177/40.11.1431064). However, adjustments in the mounting and staining protocol as well as during image capture allow for DAPI to be used in a quantitative manner (Hamada S, Fujita S. Histochemistry. 1983;79, 219–226. doi:10.1007/BF00489783). By using Fluoromount G and reduced laser intensity we followed these recommendations to avoid photobleaching and adapt DAPI staining to quantify nuclear volumes. Moreover, DAPI has been used previously to quantify the occurrence of additional endoreplications in regenerating *Drosophila* midguts (Xiang J et al. Nat Commun. 2017 May9;8:15125. doi:10.1038/ncomms15125). However, having been aware of possible inconsistencies, we used a threshold-based method rather than aggregating fluorescence intensities. The former is more robust towards variability in DAPI intercalation enabling us to precisely locate and outline nuclei in confocal z-stacks and to quantify nuclear volume as a proxy rather than DNA content itself (Figure 1C; Figure 1—figure supplement 1D,E; Figure 6C).

In parallel, we observed the occurrence of nuclear disintegration in the DPs of *Ras^V12^*-glands, which poses a two-fold problem for quantifications: (1) separating enlarged, but disintegrating from enlarged, intact nuclei in an unbiased manner and (2) separating nuclei in general due to the propensity of the DAPI-signal of disintegrating nuclei to fuse across cells and nuclei. Attempts to employ machine learning (i.e., support vector machines, k-means, Gaussian Mixture Models) to cluster disintegrating from enlarged nuclei as a prerequisite to quantify their volumes separately were not satisfactory and need more in-depth development. However, we recognized a clear separation between genotypes regarding the occurrence of nuclear disintegration, whereby *p35;Ras^V12^*-, *jnk^DN^;Ras^V12^*- and *Drs;Ras^V12^*-glands do not show any disintegrating nuclei, only *Ras^V12^*-glands (Figure 6D). Thus, we chose to show quantifications of nuclear volume as a proxy for the amount of endoreplications alongside images of DAPI stained nuclei to visualize the sole occurrence of nuclear disintegrations in *Ras^V12^*-glands (Figure 6C-D). We believe that the conclusion we drew from these two datasets is covered by the presented evidence, namely (1) by inhibiting PCD (i.e., *p35;Ras^V12^*) or inhibiting JNK signaling (i.e., *jnk^DN^;Ras^V12^*, *Drs;Ras^V12^*) nuclear disintegration is prevented, whereas (2) the same interventions do not block the induction of further endoreplications.

As indicated in our summary model, *Mmp2* is involved in tissue disintegration in as much as its secretion promotes disruption of the basement membrane, the attraction of hemocytes and thus the activation of a systemic immune response towards the hypertrophic glands (Figure 4E-G; Figure 4—figure supplement 1B-C; Author response image A-B,D; Figure 8). We agree with the reviewer that this however needed clarification throughout the manuscript.

Apart from the above-stated experimental concerns, issue with data presentation also significantly dampens the enthusiasm for this study. The data-presentation is incredibly cumbersome; throughout, the text does not present clearly the rationale of their experimental design, some key figure panels are not even mentioned in the text, other key data is buried in the supplemental figures, while other data is replicated in separate figure panels without clear reasoning, and importantly most of the legends/labeling on the graphs are so small, they are illegible. This last issue makes this work nearly impossible to critically evaluate. Also, I would encourage the authors to use gene names and symbols on their figures that are understood by a wide readership. For example, Bsk might be alternatively JNK in S5.

We have changed formatting of the figures including increasing font sizes, checked the presence of all figure panel references in the text, replaced gene names where possible (e.g., “*jnk*” instead of “bsk” as suggested by the reviewer). Supplemental figures that include data presented in the main figures show this data alongside additional genotypes in an extended format. By presenting a complete picture of all genotypes in the supplemental figures, we hope to ensure comparability amongst all datapoints.

Most importantly, we revised the manuscript’s text in order to emphasize the reason for conducted experiments as well as for the experimental set-ups.

Reviewer #3:This article is interesting because it shows that some immune genes, mostly antimicrobial peptide (AMP) genes, are induced in a highly artificial system in which the overgrowth of salivary glands is induced by ectopic expression of the Ras[V12] gain-of-function allele. The expression of immune genes would rely solely on Dorsal and not the Toll pathway (see below). The most important point is that the overexpression of Drosomycin is sufficient to reduce the induction of JNK-pathway regulated genes that occurs in the distal part of salivary glands, including pro-apoptotic genes, thus introducing the concept of a direct or indirect intracellular signaling function for an antimicrobial peptide.In the longer term, it will be interesting to determine whether Dorsal plays a role in wild-type larval salivary glands and why it is differentially repressed in the DP.Substantive concerns1) The authors use RNAi lines to determine whether the Toll pathway is required for the induction of immune genes, which are not even validated. Since it is close to impossible to demonstrate the generation of a null phenotype by RNAi, the authors should imperatively use at least one null mutant of the Toll pathway, e.g., MyD88 and possibly spz.

We have almost exclusively used RNAi-lines for the screen of Toll-/imd-signal pathway components that were previously published and found to impact the expression of the targeted gene, alter expression and have a phenotype. The corresponding references are now listed in the Supplementary file 3. However, we agree entirely with the reviewer’s comment on the necessity to validate the involvement of the Toll-pathway independently. Thus, we crossed a mutant *Myd88*-allele in a hetero- or homozygous state into the *Ras^V12^*-background (Figure 2—figure supplement 2E; Author response image 3). However, Drs-expression is not diminished at 120 h AED in the PP of *Myd88^/+^;Ras^V12^*- or *Myd88^-/-^;Ras^V12^*- compared to *Ras^V12^*-glands, whereas a *dl*-knock-out reduced Drs-expression significantly, indicating a dependence on *dl*, but not upstream Toll-signaling modules (Figure 2C; Figure 2—figure supplement 2D).

2) The authors should look at and quantify Dorsal nuclear localization in the salivary glands and not solely at its expression. Upon looking at Figure 2B, one gets the impression it is nuclear only in the very proximal region of the organ.

We agree with the reviewer’s comment on the value of quantifications based on antibody staining parallel to evaluating the expression. For this purpose, we captured z-stacks of *Ras^V12^*glands containing the DrsGFP-reporter and stained for the presence of dorsal-protein. We used DAPI to define location and dimensions of the glands nuclei and exploited the nuclear signal of DrsGFP (i.e., proxy for cellular Drs-expression) to correlate dl and Drs expression (Figure 2—figure supplement 2B; Author response image 1). Across the entire scale of quantified intensities (i.e., *log_10_*-transformed; range covers entirety from non-detectable to high intensities throughout the nucleus), dl- and Drs-signal correlate strongly with one another in both the PP and DP as shown for 5 captured glands (i.e., Pearson correlation coefficient included for all 5 samples). More importantly, dlstaining is present in both PP and DP and occurs to stimulate Drs-expression in the entire gland epithelium.

3) Multiple AMPs are induced in the PP and it is surprising that the one AMP they have overexpressed is having such a biological function. A control is missing, ideally a Drosomycin mutant in which disulfide bridges cannot form; at least a nonrelevant gene should be used as a control to exclude nonspecific effects linked to overexpression in the Ras[V12] context.

Among the induced AMPs in the acquired transcriptomes, only Drs and AttD showed considerable expression in the two sample populations derived from whole glands (i.e., *Ras^V12^*, *lgl^RNAi^;Ras^V12^*; Figure 3C; Figure 3—figure supplement 1C; Author response image 1). Given the strong AttD-induction in the PP, its expression in the whole gland might not reflect the actual distribution (Figure 3C). Thus, we performed qPCRs on separated gland parts with AttD-primers, which revealed that AttD is exclusively expressed in the PP at 96 and 120 h AED (Figure 3—figure supplement 1D; Author response image 1). According to our data, Drs is thus the only AMP that is expressed in the DP of 96 h-old *Ras^V12^*-glands. While this doesn’t exclude the possibility of other AMPs to function in a fashion similar to Drs, the latter is the only one present in the DP to alter *Ras^V12^*-induced phenotypes until 96 h AED.

In parallel, we also overexpressed CecA1, the 4^th^ highest expressed AMP in *Ras^V12^*- and *lgl^RNAi^;Ras^V12^*-glands and monitored *CecA1*-mRNA occurrence via ISH (Figure 3C; Figure 3—figure supplement 1C; Author response image 1). While CecA1 is solely expressed in the PP of 120 h-old *Ras^V12^*-glands, the employed CecA1-overepression construct driven by *Bx^MS1096^* validates the possibility for CecA1 to be expressed and detected throughout the entire salivary gland (Figure 3—figure supplement 1E; Author response image 2). However, *CecA1;Ras^V12^*-glands only exhibited CecA1-staining in the PP, indicating a *Ras^V12^*induced block of post-transcriptional mRNA-accumulation. While the precise mechanism for this block remains to be elucidated, this result is opposed to the *Drs,Ras^V12^*-coexpression, where co-expressed Drs could be detected throughout the gland (Figure 4A). This highlights the unique presence and function of Drs in the hypertrophic *Ras^V12^*-model even further.

Nevertheless, we agree strongly with the reviewer’s comment on the relevance of a control to rule out non-specific effects derived from the overexpression of the UAS-Drs construct. To mimic Drosomycin’s entry into the secretory pathway, we combined a mCD8::RFP-construct with *Ras^V12^* and repeated crucial experiments (i.e., CC3- and pJNKstaining, qPCR for Drs and *JNK*-target genes, measurement of gland size; Figure 5—figure supplement 3A-C; Figure 6—figure supplement 1C-D; Author response image 2). Co-expression of Drs with *Ras^V12^* significantly reduces the activation of Dronc and *JNK* compared with *Ras^V12^* alone, while co-expression of mCD8::RFP does not (Figure 5—figure supplement 3B; Figure 6—figure supplement 1C; Author response image 2).

The difference between *Drs,Ras^V12^*- and *RFP;;Ras^V12^*-glands is further confirmed by the performed qPCRs. Except for the *JNK*-target *Mmp2*, none of the screened genes displays significant expression differences in *RFP;; Ras^V12^*- compared to *Ras^V12^*-glands at 96 h AED (Figure 5—figure supplement 3A; Author response image 2). At the same time point, the same genes are indeed significantly downregulated in *Drs,Ras^V12^*- in contrast to *Ras^V12^*-glands. At 120 h, co-expression of Drs with *Ras^V12^*, but not mCD8::RFP, still reduces expression of Mmp1, *Mmp2* and upd1 significantly (Figure 5—figure supplement 3A; Author response image 2). In addition, *Drs,Ras^V12^*-glands show a stronger absolute reduction of *puc*-, *hid-* and *rpr-*expression than *RFP;;Ras^V12^*-glands compared to *Ras^V12^*-glands alone, in spite of a significant reduction of these genes in *RFP;;Ras^V12^*-glands.

Most importantly, while co-expression of mCD8::RFP has a moderate, but significant impact on the expression of few of the *JNK* target genes at 120 h AED, *RFP;;Ras^V12^*glands do not differ significantly in size from their *Ras^V12^*-counterparts (Figure 4—figure supplement 1A; Author response image 2). *Drs,Ras^V12^*-glands in turn are strongly overgrown indicating prolonged hypertrophic growth in the absence of *JNK*-activation and PCD. Crucially, the set of *RFP;;Ras^V12^*-control experiments indicates that Drs has a specific role in altering the physiology of *Ras^V12^*-glands that is not mimicked by merely overexpressing a mCD8::RFP-construct.

While not significant, *RFP;;Ras^V12^*-glands are indeed slightly smaller than *Ras^V12^*-glands. Since mCD8::RFP-marker constructs were to our knowledge not rigorously tested in polyploid tissues like salivary glands, it remains open as to whether mCD8::RFP itself has specific effects separate of Drs that would explain the minor reduction in tissue size. In addition, we have previously described *Ras^V12^*-induced phenotypes, while co-expressing a cytoplasmic GFP, which did not diminish the reported effects either (Figure 2 and Figure 3 in Hauling T et al. Biol Open. 2014 Apr15;3(4):250-260. doi:10.1242/bio.20146494).

Also, the Drosomycin RNAi line does not abolish the expression of its target gene and does not have such a "drastic" effect: several genes are only mildly more strongly expressed. It would be interesting to use a Drosomycin null mutant. Perhaps less important for this work but of interest, it would have been informative to test mutants affecting AMP families that have recently been published.

While we absolutely agree with the reviewer’s suggestion to use a null mutant to completely abolish Drs expression in further investigations, the *Drs^RNAi^*-line used throughout the manuscript exhibits a comprehensive reduction of Drs-expression levels. On average only 8% of the Drs-expression in *Ras^V12^*-glands remains upon *Drs^RNAi^*-co-expression at 96 h AED and only 4% at 120 h AED, in line with the virtual absence of detected *Drs*-mRNA in ISH (Figure 5—figure supplement 2A-B; Author response image 2). In addition, as exemplified for hid and puc, by knocking Drs-expression down, *puc*-expression increases by almost 50% at 96 h AED and hid-expression increases by almost 100% at the same time point (Author response image 5). As expected this increase of expression is diminished at 120 h AED (i.e., compared to *Ras^V12^*), due to the absence of Drs-expression in the distal part of *Ras^V12^*-glands, making a knock-down of *Drs* obsolete. Given the effect size of the used *Drs^RNAi^*-construct and its impact on *JNK*-target genes at 96 h AED, we believe that even without additional Drs-mutant experiments, we were able to show an effect opposite of Drs-overexpression by *Drs^RNAi^*-co-expression to strengthen the validity of our model (Figure 5—figure supplement 2C-D; Author response image 5).

The reviewer’s suggestion to use the here presented hypertrophic model to test the functionality of additional AMPs as done in 2 recently published landmark papers paves intriguing possibilities for follw-up studies (Araki et al., 2019; Parvy et al., 2019). Since Drs was the only AMP expressed in the distal part until 96 h AED, downregulated thereafter and thus anti-correlated with the activation of *JNK*-signaling, we reasoned that Drs remains the only possible candidate to mount a tissue-autonomous immune response capable of explaining the shift in phenotypes from 96 h to 120 h AED. While this does not rule out similar activities of other AMPs in this or alternative models, the focus of the presented manuscript rested upon Drs as the naturally occurring AMP in this model and its role in regulating hypertrophic growth.

4) It is not clear whether Drosomycin is secreted in the lumen or basally in the hemolymph. To really conclude that this is a tissue-autonomous response, inasmuch the authors do not state whether Drosomycin is expressed in other tissues, it would be useful to overexpress Drosomycin in the fat body and test its effects on the DP of the salivary glands.

In order to detect Drs-expression in an unambiguous manner, we employed the previously published Drs-reporter including 2450 bp of the regulatory sequence upstream of the translation start site (Ferrandon D et al. EMBO J. 1998 Aug10;17(5):121727. doi:10.1093/emboj/ 17.5.1217). Due to its systemic presence, any non-tissue autonomous induction of Drosomycin in *BxGal4>UAS-Ras^V12^*- or *BxGal4>UAS-lgl^RNAi^;UAS-Ras^V12^*-larvae would thus become obvious when monitored over time, as shown in other infection and wounding paradigms (Kenmoku H et al. Dis Model Mech. 2017 Mar1;10(3):271-281. doi:10.1242/dmm.027102; Kounatidis I et al. G3 (Bethesda). 2018;8(5):1637-1647. doi:10.1534/g3.118.200182). However, throughout larval and pupal development, a DrsGFP-signal was solely detected in the salivary glands with lymph glands, fat body, hemocytes or hematopoietic pockets all Drsnegative (Figure 1E; Figure 1—figure supplement 1E).

While we cannot unequivocally rule out interference from other tissues, the most parsimonious explanation based on the presented evidence is a tissue-autonomous expression and activity of Drs in the hypertrophic glands: (1) Expression and presence of Drs in a spatiotemporal pattern that precisely anti-correlates with the induction of *JNK*-signaling. (2) Rescue of tissue degradation in the distal part by Drs-overexpression across the gland’s epithelium at 120 h AED. (3) Further increase of *JNK*-activation upon tissue-wide knock-down of Drs at 96 h AED. (4) Physiological expression of dl as a crucial upstream activator of Drs in a spatiotemporal pattern completely overlapping with Drs. (5) Absence of any non-tissue-autonomous expression of Drs, especially no other dedicated immune organ.

We very much appreciate the reviewer’s suggestion to test further inter-organ communication between the hypertrophic glands and amongst others the fat body and agree that further evaluations would be of interest. However, in the light of substantial evidence for a tissue-autonomous activity of Drs, we deem establishing, optimizing and validating a dualexpression system (i.e., 2 two-component expression systems, QF-QUAS or LexA-lexAop alongside Gal4-UAS) for parallel expression of *Ras^V12^* in the salivary glands and Drs in the fat body exceptional for an evaluating experiment.